# Comparative Analysis of Three Predictive Models of Performance Indicators with Results-Based Management: Cancer Data Statistics in a National Institute of Health

**DOI:** 10.3390/cancers15184649

**Published:** 2023-09-20

**Authors:** Joel Martínez-Salazar, Filiberto Toledano-Toledano

**Affiliations:** 1Unidad de Investigación en Medicina Basada en Evidencias, Hospital Infantil de México Federico Gómez, Instituto Nacional de Salud, Dr. Márquez 162, Doctores, Cuauhtémoc, Mexico City 06720, Mexico; jmartinez@himfg.edu.mx; 2Unidad de Investigación Multidisciplinaria en Salud, Instituto Nacional de Rehabilitación Luis Guillermo Ibarra Ibarra, Calzada México-Xochimilco 289, Arenal de Guadalupe, Tlalpan, Mexico City 14389, Mexico; 3Dirección de Investigación y Diseminación del Conocimiento, Instituto Nacional de Ciencias e Innovación para la Formación de Comunidad Científica, INDEHUS, Periférico Sur 4860, Arenal de Guadalupe, Tlalpan, Mexico City 14389, Mexico

**Keywords:** results-based management, performance indicators, predictive models, evidence-based decision making, cancer data statistics

## Abstract

**Simple Summary:**

Statistical predictive models using one of the most important strategies, known as results-based management (RBM), are relevant for improving the quality of medical services and could be used with cancer data statistics to monitor and evaluate children with cancer. We provided a comparative analysis of three predictive models that are considered robust in the literature. We also provided a tool to identify new medical cases that confirm a forecast increase or decrease in certain performance indicators and can convert data into useful information for decision-making; we propose to apply this approach to cancer data statistics at one of the National Institutes of Health in Mexico City. Our findings show that from an RBM-based perspective, predictive models are a valid and reliable instrument to forecast medical performance indicator results and can be applied to monitor and evaluate children with cancer.

**Abstract:**

Predictive models play a crucial role in RBMs to analyze performance indicator results to manage unexpected events and make timely decisions to resolve them. Their use in Mexico is deficient, and monitoring and evaluation are among the weakest pillars of the model. In response to these needs, the aim of this study was to perform a comparative analysis of three predictive models to analyze 10 medical performance indicators and cancer data related to children with cancer. To accomplish these purposes, a comparative and retrospective study with nonprobabilistic convenience sampling was conducted. The predictive models were exponential smoothing, autoregressive integrated moving average, and linear regression. The lowest mean absolute error was used to identify the best model. Linear regression performed best regarding nine of the ten indicators, with seven showing *p* < 0.05. Three of their assumptions were checked using the Shapiro–Wilk, Cook’s distance, and Breusch–Pagan tests. Predictive models with RBM are a valid and relevant instrument for monitoring and evaluating performance indicator results to support forecasting and decision-making based on evidence and must be promoted for use with cancer data statistics. The place numbers obtained by cancer disease inside the main causes of death, morbidity and hospital outpatients in a National Institute of Health were presented as evidence of the importance of implementing performance indicators associated with children with cancer.

## 1. Introduction

### 1.1. Results-Based Management to Address the Problem of Improving Public Health Services

Results-based management (RBM) is a public management strategy that aims to improve the efficiency and effectiveness of policies, programs and public services based on reliable information about the effects of governmental actions on society. RBM was introduced in response to the increasing demands and expectations of citizens, such as universal coverage of education and health services, the expansion of the rights of marginalized populations, and concerns about sustainable management of the environment. RBM provides a reference framework to facilitate public organizations in creating public value (results) [1].

The RBM model is used in Countries of the Organizations for Economic Cooperation and Development, the European Community, Latin America, the Middle East, Africa, and Asia. However, its use in Mexico is deficient or incorrect. Since the Paris Declaration in 2005, numerous governments and organizations worldwide have documented the effectiveness of the RBM approach in achieving significant changes by measuring the impact of public sector activity on people’s quality of life. The literature shows how different agencies worldwide strive for greater effectiveness and efficiency in developing their countries [2].

According to Dwivedi et al. [3], the COVID-19 pandemic has challenged the capacity of health management to make timely decisions and effectively manage such events. The evaluation of an organization is based on performance indicators, and analyzing these indicators provides insight into the impact on the quality of services they provide [3].

### 1.2. Performance Indicators and Predictive Models for Measuring Healthcare Results with RBM

Performance indicators are mathematical expressions used to measure quantitative or qualitative variables that reflect changes linked to actions, enabling the monitoring and evaluation of results. These indicators are measured in units such as proportions, percentages, averages, rates, and indices [4].

An organization’s results are measured in four dimensions with RBM: 1. Effectiveness; 2. Quality; 3. Economy; and 4. Efficiency. Effectiveness measures the level of achievement of objectives. Quality measures service attributes and compliance with requirements. Economy measures the capacity to adequately generate and allocate financial, human, and material resources to the goods or services provided. Efficiency measures the resources used compared to the results obtained [4].

Performance is a strategic concept in which the goals achieved are associated with the resources used to achieve them [5]. The Mexican government’s entities and dependencies must demonstrate by law that public spending benefits society and promotes adequate accountability and transparency [6].

The hospital’s performance is associated with the results of performance indicators, which can be analyzed and evaluated to assess their ability to achieve goals efficiently. This evaluation can be used to improve the positive impact of health services [7]. These findings agree with other research that found that performance indicators and statistical predictive models are important to measure a health system’s quality [8].

There are two fundamental processes in performance indicator measurement: 1. Monitoring during the execution of activities and tasks carried out by organizations and 2. Performance evaluation at the end of public programs and projects. The first process involves follow-up on the achievement of objectives and goals [9]. Performance evaluation with RBM can be conducted in terms of design, processes, consistency, and results. Its function is to provide reliable information that helps to make better decisions for continuous improvement processes [10].

#### 1.2.1. Predictive Models for Measuring Healthcare Results

Predictive models play a crucial role in public decision-making, particularly in the context of local and regional policies. Performance indicator values are typically arranged in time series data to generate accurate predictions that enable effective resource planning and optimization [11].

#### 1.2.2. ARIMA Model

The ARIMA model was selected for this study due to its ability to predict results with autoregressive (AR) and moving average (MA) components when modeling changes in performance indicators over time. The ARIMA integration (I) component is added to determine the level of differentiation required to ensure data stationarity [12] and is effective in capturing linear patterns in time series data [13].

Sun et al. [14], Afilal et al. [15], and Milner [16] developed ARIMA forecasting models to predict patient attendance at hospital services and validated the model as an effective tool for predicting the number of expected annual patients in the Trent region of the United Kingdom, with a forecast error of only 3 percent. In a similar study, Sun et al. [14] were able to enhance their predictions of daily patient attendance at Singapore General Hospital by integrating other variables, including public holidays and frequency of appointments, into their ARIMA predictive model.

#### 1.2.3. Exponential Smoothing (ES) Model

ES is the most common forecasting technique and is widely used by businesses of all kinds, including healthcare. ES techniques became widespread for six main reasons [17]:Exponential models are surprisingly accurate.Formulating an exponential model is relatively easy.The user understands how the model works.Very few calculations are required to use the model.Computer storage requirements are low due to the limited use of historical data.Accuracy tests related to model performance are easy to calculate.

ES forecast models are commonly used for time series forecasting due to their robustness, particularly for short periods with limited historical data. These models have been utilized in various fields to generate accurate predictions, with different studies highlighting their efficacy [18].

Kotillová [19] utilized performance indicator results to predict electricity demand in Australia and discovered that ES models produced highly accurate predictions with minimal errors. Similarly, Köppelová and Jindrová [20] found that ES performs well in terms of error and provides good results.

#### 1.2.4. Linear Regression Model (LR)

LR is probably the most widely applied in both natural and social sciences research, as well as in daily life [21]. Regression analysis is defined as ‘the study of dependency’ or how a result depends on one or more predictors or independent variables [22].

LR and correlation are powerful statistical tools that can serve as effective predictive models when used properly. However, improper use can lead to meaningless results. Three examples of the inappropriate use of LR are as follows: 1. LR should not be used when the relationship between the dependent and independent variables is nonlinear. If the data exhibits a curved or nonlinear pattern, it may lead to inaccurate predictions and incorrect conclusions. To test the null hypothesis that a given dataset follows a normal distribution, the Shapiro–Wilk test could be used, if the *p*-value > 0.05 then we fail to reject the null hypothesis, suggesting that the data is normally distributed [23]. 2. LR should check the homoskedasticity assumption, which is a desirable property where the variability of the errors (residuals) is constant across all levels of the independent variable. One way to detect heteroskedasticity is by using the Breusch–Pagan test, the null hypothesis is that there is homoskedasticity in a LR model. If the *p*-value > 0.05, we fail to reject the null hypothesis, suggesting homoskedasticity. The problem with heteroskedasticity is that it may lead to incorrect estimates of the standard errors for the coefficients and, consequently obtain incorrect t values [24]. 3. Improperly handling outliers can unduly influence the regression line and impact the accuracy of the model. LR is sensitive to outliers, which are data points that deviate significantly from the overall pattern. Cook’s distance (Di) can be used in regression analysis to identify influential outliers in a set of predictor variables. Large values (usually greater than 1) indicate substantial influence by the case in affecting the estimated regression coefficients [25].

LR is applied in numerous areas: medicine, engineering, economics, and social sciences, among many others. It is an expansive field that has gained significant interest among researchers and administrators [26,27,28].

### 1.3. Other Management Models for Measuring Healthcare Performance

Different models have been developed in addition to RBM, such as data envelopment analysis (DEA) [29], Pabon lasso [30], analytical hierarchy process (AHP) [31], and balanced scorecards (BSC) [32].

The DEA model uses nonparametric deterministic analysis as an alternative to other efficiency measurement techniques, its complexity makes it difficult for many hospital managers [33,34].

The Pabon lasso model was introduced in 1984 for the identification of the state of a hospital in terms of resource utilization and performance trends.

The AHP model involves breaking down a complex decision into a hierarchy of smaller, more manageable subdecisions and then assigning relative weights criteria based on the importance [35]. The approach breaks down a complex decision problem suggested by Dolan et al. [36]: 1. Define the decision criteria and alternatives, 2. Rate the criteria by pairwise comparison, 3. Calculate the weights of priority, 4. Calculate the overall priority of the alternatives, 5. Check for inconsistencies and 6. Perform a sensitivity analysis [37].

The BSC model uses analysis of performance indicators associated with four perspectives: finance, internal processes, customers, development and research. Three studies suggest the use of many statistical methods, such as the mean, standard deviation, coefficients of variation, LR, Student’s *t*-test, covariance analysis, analytic data, confirmatory factor analysis (CFA), and structural equation modeling (SEM). The latter two use the following indices: goodness-of-fit index (GFI), modified goodness-of-fit index (AGFI), comparative fit index (CFI), normalized fit index (NFI), index of relative fit (RFI), incremental fit index (IFI), Tucker Lewis index (TLI), residual root mean square (RMR), residual root mean square approximation (RMSEA), chi-square, and significance level [38,39,40]. To clarify some of the mentioned statistical methods, structural equation modeling (SEM) is a technique used in behavioral sciences that can be applied within the BSC model to comprehend and make decisions related to marketing and customer behavior. SEM focuses on theoretical constructs represented by latent factors, and the relationships between them are depicted through regression or path coefficients. It establishes a covariance structure among observed variables, providing an alternative modeling approach. Confirmatory factor analysis (CFA) is used to assess the fit of the model and determine whether the underlying structure of a scale aligns with the validity of self-report measures, such as surveys, questionnaires, interviews, and psychometric tests. The CFA process involves specifying a factor structure based on the theoretical construct and estimating factor loadings, which represent the correlations between the observed items and the underlying factors. Fit indices, such as the Comparative Fit Index (CFI), Tucker–Lewis Index (TLI), Root Mean Square Error of Approximation (RMSEA), and Standardized Root Mean Square Residual (SRMR), are then used to evaluate how well the model fits the observed data. A well-fitting model will have CFI and TLI values close to 1 and low values for RMSEA and SRMR, thus enabling informed decision-making [41]. Importantly, statistical methods can be applied to any existing management model, not limited to the BSC.

#### Other Models with Statistical Methods to Measure Healthcare Results

Analytics data refer to different analyses that can be used with performance indicator values, including (i) descriptive analysis; (ii) operational analysis; (iii) online analytical processing (OLAP); (iv) statistical analysis; (v) data mining, predictive models; (vi) visual analytics, such as dashboards and statistical graphs; (vii) text mining as a natural language process; (viii) big data analytics (BDA); (ix) network analysis (x) web mining; (xi) sentiment analysis; and (xii) analysis of social networks [42,43,44,45].

Business intelligence (BI) refers to the use of methods to support the decision-making process by transforming raw data into useful information [46,47]. The literature review revealed that the combination of business intelligence (BI) and data analytics is a promising technique for evaluating the management of hospital facilities. However, also identified certain limitations and recommended further research in this area [48].

Another study focused on data mining (DM), to generate useful information from databases. Ahmed et al. [49] and Fan et al. [50] stated that DM can be classified under supervised learning and unsupervised learning by learning machines. While supervised learning aims to produce predictions using data, unsupervised learning seeks to discover new insights from aggregated historical data. Ioannidis et al. [51] and Yafooz et al. [52] mentioned five categories of data mining: 1. anomaly detection or outlier detection; 2. association rule learning; 3. clustering analysis; 4. classification and analysis; and 5. regression analysis.

Hospitals are the most important cost factor in health systems worldwide. The health sector is associated with people’s lives and with the highest levels of complexity in business models [53].

One limitation found was incorrect forecasting methods and a lack of understanding or confusion about the methodology. As a result, organizations often set goals and objectives without realistic forecasts to justify the probability of achieving them [54]. A Systematic Review of Financial Literacy Research in Latin America and The Caribbean mentioned that one of the most common problems is the lack of statistical test validity and reliability, it is necessary to improve the quality of these models [55]. The literature shows that the DEA model has deficiencies regarding the qualitative and methodological approaches used [29]. The Pabon lasso model showed no significant changes in its results in a performance evaluation of public general hospitals in Yugoslavia [30]. In the AHP model, as in the DEA model, performance indicator values are used less frequently compared to other models, and their use in the healthcare area is reported to be inconsistent. Finally, a systematic review of the BSC model found that articles mainly focus on its design process rather than its implementation, use and results [32].

One limitation of RBM in Latin America and the Caribbean is that monitoring and evaluation with predictive methods are among the weakest pillars of the model. In Mexico, studies have shown that monitoring and evaluating results using RBM and performance indicators associated with the National Institutes of Health lacked methodological rigor to be useful and credible [56]. Another study on RBM model implementation in Mexico City from 2018 to 2021 found a poor understanding of the methodology. No evidence of the use of predictive methods to analyze performance indicators for evidence-based decision-making and to enhance the quality of public services was found in the literature [57].

### 1.4. Problem Statement

Research regarding the impact of RBM supported by predictive models on healthcare performance indicators is promising, but one limitation is the availability of reliable measurement statistical methods and the fact that they must be validated and associated with cancer data statistics in a National Institute of Health. The comparative analysis of 3 predictive models can be useful for this purpose and has the advantage of having been developed in the Mexican cultural context. However, more complex models do not necessarily predict real values more accurately, and the model that fits the data best depends on the medical process context in which the results are obtained.

In response to these needs, the aim of this study was to perform a comparative analysis of three predictive models considered robust in the literature to analyze 10 medical performance indicators and cancer data statistics related to children with cancer. The predictive models used were ARIMA, LR and ES. We considered three objectives: (1) identify the predictive model that obtains the smallest error between the real and predictive values to be used as RBM support; (2) describe how predictive models can be used as acceptable tools to analyze the results of performance indicators associated with RBMs; and (3) prove the validity of the RBM in relation to the performance indicator results analysis with predictive models in a National Institute of Health in Mexico City.

## 2. Materials and Methods

### 2.1. Design

A comparative and retrospective study with nonprobabilistic convenience sampling was conducted with 10 medical performance indicator results to evaluate the performance of the Hospital Infantil de Mexico Federico Gómez which is one of the National Institutes of Health in Mexico, and a number of patients and cancer diagnostics were included (Table 1). The inclusion criteria for performance indicators in this study were as follows: medical attention services; results obtained in the period from January 2014 to December 2018; and the existence of results in the analyzed period. Data were collected and analyzed by the authors of this paper from the quarterly historical reports used at the evaluation performance meeting that was held according to hospital procedures.

### 2.2. Procedure for Measuring the Performance of Each Model

Time series containing at least 20 data points were used for prediction. We made quarterly forecasts of performance indicator results with four quarterly lagged results as a predictor. The fitting and forecasting performances of the 3 models were evaluated (ARIMA, LR and ES)

The performance criterion used to test the predictive accuracy was the mean absolute error (MAE) of the 3 statistical predictive methods. A low MAE indicates a good fit and was calculated as follows:$$MAE=\frac{1}{n}\sum_{i=1}^{n}\left|Ei\right|,$$
where *Ei* is the error or difference between the real and predicted values.

### 2.3. Predictive Statistical Methods Used to Measure Performance Results

#### 2.3.1. Linear Regression

The first step in LR is to determine the slope or inclination of the straight line, whose algebraic representation is as follows [21]:
E (Y/X) = a + bx,
where E is the estimated or predicted value of “Y” given a value of “X” that is equal to a + b multiplied by “x”, assuming that the distribution of “Y” for a value of “x” follows a normal distribution and that the variances of both variables are homogeneous, a phenomenon known as homoscedasticity.

The most popular way to mathematically represent LR is as follows:
Y = α + β X + e,
where:

Y is the explained or dependent variable that is the estimated or predicted value based on a sample.

α is the value at which the line intersects the *Y*-axis, equivalent to the value when x is equal to zero, also known as the regression constant.

β is the value that measures the influence that explanatory variable X has on the explained or dependent variable Y.

e is the error. When the equation is solved, it results in:

e *=* Y − (α + β X),

Thus, e indicates the amount by which Y deviates from the obtained values.

#### 2.3.2. ARIMA

Figure 1 shows the ARIMA(p,d,q) model; for nonstationary time series data, the mean and variance are unstable, and such data are generally converted to a stationary time series first by means of a differential operation [58].

The ARIMA model integrates autoregressive and moving average calculations. A basic requirement for predicting time series with ARIMA is that the time series is stationary or, at the very least, trend stationary [16,59]. ARIMA expects a stationary stochastic process as input; however, very few datasets are natively in such a format; thus, the use of differencing to “stationary” is performed in the model identification stage [60].

#### 2.3.3. Stationary

A stationary time series is one whose properties do not depend on the time at which the series is observed. Time plots will show the series to be roughly horizontal (although some cyclic behavior is possible), with constant variance [61].

#### 2.3.4. Differencing

Computing the differences between consecutive observations is known as differencing, which is one way to make a nonstationary time series stationary. Differencing can help stabilize the mean of a time series by removing changes and therefore eliminating (or reducing) trends and seasonality. For a stationary time series, the ACF will drop to zero relatively quickly, while the ACF of nonstationary data decreases slowly. Additionally, for nonstationary data, the value of *r_1_* is often large and positive (Figure 2).

#### 2.3.5. Autocorrelation

Just as correlation measures the extent of a linear relationship between two variables, autocorrelation measures the linear relationship between lagged values of a time series. There are several autocorrelation coefficients corresponding to each panel in the lag plot. For example, *r_1_* measures the relationship between y_t_ and y_t_ − 1, *r_2_* measures the relationship between y_t_ and y_t_ − 2, and so on (Figure 2).

The value of rk can be written as
$$\frac{\sum_{t=k+1}^{T}\left(y_{t}-\hat{y}\right)\left(y_{t-k}-\hat{y}\right)}{\sum_{t=1}^{T}{\left(y_{t}-\hat{y}\right)}^{2}},$$
where ŷ is the mean of the historical data and *T* is the length of the time series. Figure 2 shows scatterplots of nine values. The autocorrelation coefficients are plotted to show the autocorrelation function or ACF. This plot is also known as a correlogram.

In this graph:*r*_4_ is higher than the values for the other lags. This is due to the seasonal pattern in the data: the peaks tend to be four quarters apart, and the troughs tend to be two quarters apart.*r*_2_ is more negative than the values for the other lags because troughs tend to be two quarters behind peaks.The dashed blue lines indicate whether the correlations are significantly different from zero.

#### 2.3.6. White Noise

Time series that show no autocorrelation are called white noise. Figure 3 gives an example of a white noise series.

For the white noise series, we expect each autocorrelation to be close to zero, 95% of the spikes in the ACF should lie within ±2/$\sqrt{T}$ where T is the length of the time series. It is common to plot these bounds on an ACF graph (the blue dashed lines above). If one or more large spikes are outside these bounds or if substantially more than 5%, then the series is probably not white noise.

#### 2.3.7. Autoregressive AR(p)

This component is a multiple regression that forecasts the variable of interest using a linear combination of its past values. It is called autoregression because it is a regression against the variable itself.

The model can be written as:
*y_t =_ C + Փ*_1_*y_t_*_−1_ + *Փ*_2_
*y_t_*_−2_ +…+ *Փ_p_ y_t-p_* + *ε_t_*,
where *εt* is white noise, and the AR(p) model is the autoregressive model of order *p* for multiple regression in period *t* with predictive values of *yt*.

#### 2.3.8. Moving Average component MA(q)

Instead of using past values of the variable to be forecasted in a regression, the moving average uses past forecast errors in a model similar to regression.
*y_t =_ C + ε_t_ +θ*_1_*ε_t_*_−1_ +*θ*_2_*ε_t_*_−2_ + … + *θ_q_ε_t-q_*,
where εt is white noise. We refer to this as an MA(q) or a moving average component of order q. Of course, we do not observe the values of εt, so it is not truly a regression in the usual sense. Moving average should not be confused with moving average smoothing. A moving average is used to forecast future values while moving average smoothing estimates the trend cycle of past values.

To define a series of values as nonstationary, the backward shift operator (*B)* is introduced. The time series *yt* is homogeneous if it is nonstationary and its first or dth difference produces a stationary time series, which means that *w_t_* = *y_t_* − *y_t_*_−1_ = (1 − *B*)*y_t_*, o *w_t_*$={\left(1-B\right)}^{d}$*y_t_*. The model is known as ARIMA(p,d,q) if its d-th difference results in a stationary process and can be written as:
$$Փ(B)(1-B{)}^{d}yt_{\;}=\delta +Ө(B{)}^{d}\varepsilon_{t},$$where
$$Փ(B)=1-\;\sum_{1=1}^{p}Փi\;{B^{i}}_{\;(1)\;\;and},$$
$$Ө(B)=1-\;\sum_{1=1}^{q}\theta i\;B^{i\;(2)},$$

B^i^
_(1) and_,

The lag terms in the autoregressive AR(p) and moving average MA(q) models are defined as follows:
Փ(B) y_t_ = δ + ε_t_,
y_t_ = µ + Ө(B) ε_t_, con δ = µ − Փ µ,
where µ is mean and ε_t_ is white noise, whose expected value is zero, E(ε_t_) = 0. The orders of the model p and q are determined by the autocorrelation and partial autocorrelation functions using the Box and Jenkins method [14]. The best model is identified based on the Akaike information criterion (AIC), the Bayesian information criterion (BIC), and the Jarque–Bera normality test on the residual error series.

The Dickey–Fuller test is used to determine how many differences are needed for the time series to present stationary, if the null hypothesis of having a unit root can be rejected (i.e., if they are nonstationary), favoring the alternative hypothesis that the data have no unit root and are therefore stationary.

Often, the only value that does not achieve significance is the trend, in which case it is possible to use the ARIMA statistical function with the “noconstant” parameter, indicating that the value of the trend is not significant, if the function yields significant data for the number of autoregressive and moving average terms tested, the parameter d can be eliminated, and the model is used with only two components ARMA(p,q).

If the null hypothesis rejection cannot be supported with one difference, its value is increased by one, and the test is repeated until the difference produces significant data; there is a possibility of not finding stationarity, in which case the d value of the difference takes a value of zero and the model becomes ARMA because it lacks the integrating stationary component.

The number of autoregressive AR(p) values recommended for use is found via the graphical function of autocorrelation, the number of values observed outside the confidence interval or close to its limit are the ones to be tested to find the ARIMA function that generates significant statistical values.

#### 2.3.9. Exponential Smoothing

This model searches for an optimal factor α in (0, 1) that best fits the actual results obtained. ES needs only three data points to forecast the future: the most recent forecast, the actual value in the forecast period, and a smoothing constant alpha (α). The smoothing constant determines the level of uniformity and the speed of reaction to differences between forecasted results and actual results. The equation for ES forecast is simply [17]:
*Ft = Ft−1 + α* (*At−1 − Ft−1*),
where:

Ft = Exponentially smoothed forecast for period t

Ft−1 = Exponentially smoothed forecast for the previous period

At−1 = Actual demand in the previous period

α = Desired responsiveness index or smoothing constant.

In many cases, the most recent facts are more indicative of the future than are those from the more distant past. If this premise is valid (that the importance of data decreases as the past becomes more distant), it is likely that the most logical and easiest method is ES. The reason it is called exponential smoothing is that each increment in the past is reduced by a factor of (1 − α).

### 2.4. Ethical Considerations

This study was conducted in accordance with research protocol HIM-2018-053, which was approved by the Research, Ethics, and Biosafety committees at the Hospital Infantil de México Federico Gómez National Institute of Health. As noted, the subject studies are performance indicator results and correspond to public information according to Mexican transparency law [62].

### 2.5. Statistical Analysis

The database was built using Excel 2016, and statistical analysis was completed using STATA version 12 (Stata Corp; Lakeway, TX, USA) software. The number of patients and place numbers obtained by cancer disease inside the main causes of death, morbidity and hospital outpatient and their percentage was presented as statistical evidence (Table 1), of the importance of implementing performance indicators associated with cancer to use RBM and predictive models in the monitoring and evaluation of children with cancer.

For the first specific objective, the MAE result for each model was used to identify the predictive model with the smallest error between the real and predictive values. The smallest standard deviation also was considered as an indication of data consistency and homogeneity, as it resulted in better predictive values in relation to the mean.

For the second specific objective to describe how predictive models can be used as acceptable tools to analyze the performance indicator results, statistical analysis for each model was conducted as follows:

A value of *p* < 0.05 is used to provide evidence to reject the null hypothesis in favor of the alternative hypothesis. For the ARIMA method, the Dickey–Fuller test was employed with the null hypothesis that time series data has a unit root equal to 1 or it is non-stationary. The alternative hypothesis asserts that the root differs from 1, signifying the absence of a unit root and ensuring that the data is stationary with a stable mean and variance. Additionally, the MacKinnon test was utilized for its enhanced precision in critical values, particularly for small sample sizes. This test represents an improved iteration of the Dickey–Fuller test and aligns with the aforementioned logic.

If the data were not stationary, autocorrelation (AC) and partial autocorrelation functions (PAC) were used to assess the difference until the data became stationary to be used as the AR(d) component. The lag number observed with the PAC function was used as the AR(p) component, and the number of values observed outside the confidence interval or close to its limit were the ones to be tested. The autoregressive number was calculated in an incremental process to establish the AR(q) component. Finally, the ARIMA model was applied based on the three components observed.

The second criterion to consider the ARIMA(p,d,q) model acceptable is to obtain a significance *p*-value for each parameter.

For the LR model the null hypothesis is that the coefficient of the independent variable (time) is equal to zero, meaning that it does not have a significant effect on the dependent variable (performance indicator results). The alternative hypothesis, in contrast, asserts that there is a significant linear relationship between the variables. A significant *p*-value in linear regression means that there is sufficient evidence to reject the null hypothesis and suggests that there is a relationship between the independent variable and the dependent variable.

For the ES model, the “desired responsiveness index or optimal factor α” was calculated to obtain the predicted values. ES is not directly associated with a *p*-value as in statistical hypothesis testing. Instead, exponential smoothing is a forecasting method that uses a weighted averaging approach to predict future values in a time series.

For the third specific objective to prove the validity of the RBM in relation to the performance indicator results analysis with predictive models, the best model can be used in order to calculate future expected values.

## 3. Results

### 3.1. Cancer Data Statistics

Table 1 shows the place number and percentage of cancer in terms of the cause of death, cause of morbidity and hospital outpatients in the period from 2016 to 2021. The year 2020 does not appear due to the activity alterations produced by the COVID-19 pandemic. Notably, the main causes of death in 2017, 2019 and 2021 were tumors and neoplasms. With respect to the main cause of morbidity, acute lymphoblastic leukemia was the most common disease in 2016, 2017, 2018 and 2021. Finally, the main cause of hospital outpatient visits in the five years was tumors and neoplasms.

### 3.2. The Best Predictive Model That Fit the Data with the Lowest Error

Table 2 shows summary statistics of the predictive models. The lowest MAE and standard deviation indicate the best fit of the data, and the *p*-value provides evidence of the significance reached.

### 3.3. Predictive Model Results

#### 3.3.1. ARIMA

Table 3 for the ARIMA criteria results for each performance indicator’s results indicates that none obtain significance for the three components of value, trend, and constant used to make the data stationary. The results of the indicators that best fit the ARIMA model are three that only obtain a nonsignificant value in their trend, and this can be corrected with the “noconstant” option of the ARIMA function: bed occupancy rate, bed turnover rate, and bed turnover index. The integration component or number of differences required to achieve stationary data was zero for the three mentioned indicators, which means that the models used were ARMA (autoregressive with moving averages without integration).

Table 3 shows that the models with the lowest MAE, resulting from the difference between the real value of the indicator and the estimated value, did not result in the lowest difference in any of the performance indicators’ results for ARIMA. An exception is the indicator of hospital admissions through emergencies, where the predictive error and its standard deviation are minimal; however, the *p*-value did not reach significance.

#### 3.3.2. Linear Regression

Seven indicators show better adjusted results with the LR model by showing lower predictive error and *p*-values < 0.05: 1. Bed Occupancy rate; 2. Hospital admissions through the emergency department; 3. Bed turnover rate; 4. Bed rotation index; 5. Percentage of emergency admissions; 6. Major surgery index; and 7. Proportion of subsequent consults with relation to the first time (Table 2).

Table 2 shows the remaining three indicators do not reach a *p*-value < 0.05 with the LR model; however, their predictive error is the lowest of the three models that were compared. The assumptions were checked as shown in Table 4.

Seven performance indicators demonstrate normality as per the Shapiro–Wilk test. However, the first two indicators and the percentage of emergency admissions do not exhibit a normal distribution. It is important to note that since *p* > 0.05, we do not reject the null hypothesis suggesting that the data follows a normal distribution. We have effectively addressed outliers; within the 20 data points analyzed, three indicators have one or two data points exceeding the threshold. This indicates that the simple linear regression model is not significantly affected by unusual values. Lastly, the Breusch–Pagan test reveals that nine out of the ten indicators show homoskedasticity, confirming the accurate estimation of coefficient errors. Similarly, as with the Shapiro–Wilk test, the *p*-value > 0.05 supports that the null hypothesis could not be rejected suggesting homoskedasticity.

#### 3.3.3. Exponential Smoothing (Table 2)

Three indicators show an acceptable fit to the ES model: 1. Mortality rate; 2. Percentage of scheduled consults granted; and 3. First-time consults index.

## 4. Discussion

The aim of the present study was to perform a comparative analysis of three predictive models considered robust in the literature to analyze 10 medical performance indicators and cancer data statistics related to children with cancer. The findings show that tumors and neoplasms were the main causes of death in 2017, 2019 and 2021; acute lymphoblastic leukemia was the most common disease in 2016, 2017, 2018 and 2021; and finally, the main causes of hospital outpatient visits in the five study years were tumors and neoplasms. These results justify the importance of implementing performance indicators associated with cancer to use RBM and predictive models in the monitoring and evaluation of children with cancer.

The first specific objective of this research was to identify the predictive model that achieves the smallest error between the real and predicted values. Three statistical predictive models were used and described to illustrate their use to help administrators, professionals, and decision-makers convert data into useful information for decision-making purposes [5] at the National Institute of Health. The performance indicator results could be analyzed with the statistical predictive models selected to help RBM identify which best fit the real values, so it could be used to address the goals and targets established during the planning process using values based on statistical evidence to obtain a realistic forecast to justify the probability of achieving them [54] and could also be used to analyze cancer data statistics.

The forecast values obtained with predictive methods are useful to monitor the results, to identify any value not expected to review what happened in the process and to take actions to correct errors or improve the effectiveness and quality of medical services delivered [53]. This approach increases the reliability of the monitoring and evaluation performance process, which was identified as one of the weakest pillars of the RBM model in Latin America and the Caribbean. Furthermore, other National Institutes of Health in Mexico could promote its use to increase RBM methodological rigor and credibility [55].

The results show that more complex models do not necessarily predict real values more accurately and prove that RBM with statistical methods could be used as a demonstration tool for the effectiveness and efficiency of hospitals’ performance [2] and their future trends.

The second specific objective of the study was to describe how predictive models can be used as acceptable tools to analyze the results of performance indicators associated with RBM. The expectation was described step by step for each predictive model and the criteria used to determine the acceptance of the predicted results. This research addressed such tasks and tried to provide an understanding and avoid confusion about the methodology used to search for accurate results regarding the hospital’s performance. Moreover, guidelines could be formulated to encourage their use at the Hospital Infantil de Mexico Federico Gomez with an improved capacity for result management with RBM [55]. To check the method’s assumptions of the LR model, we employed the Shapiro–Wilk, Cook’s distance, and Breusch–Pagan tests. Our results demonstrate that the model fits appropriately for the majority of the performance indicators. The adequate quality of the predictive statistical methods to analyze performance indicator results is expected to improve RBM for use with cancer data statistics.

Our findings related to the second specific objective strengthen the predictive methods, consolidate them and overcome the limitations found in Latin America and the Caribbean and the lack of methodological rigor in the National Institutes of Health to ensure credibility and their usability to monitor performance indicator results and analyze statistics associated with children with cancer [56]. The set of criteria identified to accept a predictive model as a tool that can be used to analyze performance indicator values provides several factors to understand the methodology and help the National Institute of Health improve the knowledge of RBM and decision-making based on evidence about public spending to enhance the quality of healthcare [57]. All the previously mentioned results provide motivation that justifies the proposal to use predictive methods with RBM for the analysis of cancer data statistics, and importantly, performance indicators focused on cancer data statistics do not exist in the National Institute of Health, where the research was conducted. Therefore, this study notes the importance of designing and implementing such performance indicators to benefit children with cancer.

The results are congruent with the importance that Mexican government law assigns to increasing country development and ensuring accountability and transparency [6]. The four performance indicator dimensions are better understood and used to push effectiveness, quality, economy and efficiency [4]. Cancer data statistics must be included to evaluate the medical treatment of children with cancer and must be reflected in the place occupied by cancer as the main cause of death, morbidity and hospital outpatients.

The third objective of the study was to prove the validity of the RBM in relation to the performance indicator results analysis with predictive models in the National Institute of Health in Mexico City. The expectation was that the validity of a predictive model depends on statistical significance, which was demonstrated with error calculations and with the above step-by-step description. The study provides a better understanding of the methodology and strengthens the knowledge about the model. The use of predictive models to analyze performance indicators increases decision-making based on evidence and could be promoted to support cancer data statistics and public policies in Mexico’s health sector [57].

The validated model can be used with financial performance indicators to associate the cost factor with medical treatment and with the highest levels of hospital complexity as a business model [53].

It is worth noting that the study has several limitations. Firstly, the absence of probabilistic sampling impacts the generalizability of findings. Secondly, the utilization of *p*-values for decision-making warrants caution due to their sensitivity to sample size; in large samples, *p*-values may appear small even when differences are absent, and vice versa. Additionally, minor fluctuations in data or analytical methodologies can lead to substantial variations in *p*-values, influencing result interpretations.

Moreover, the tests chosen to assess LR assumptions (Shapiro–Wilk, Cook’s distance, and Breusch–Pagan tests) might not be the most suitable options and could benefit from the inclusion of Q-Q plots. Tests relying on *p*-values can sometimes detect insignificant differences as significant in larger sample sizes, while the reverse problem arises for smaller samples. Notably, there are other statistical tests available that were not utilized to assess LR assumptions, such as assessing the independence of errors, detecting multicollinearity, identifying autocorrelation, and verifying constant variance. Additionally, the three statistical predictive models were designed to follow linear function data; consequently, the forecast values that were found are limited to the linear behavior of the statistical method used.

Future studies must include nonparametric methods to account for nonlinear function behavior; it is also important to consider the use of machine learning statistical methods and informatics tools to collect, analyze and present the performance indicator results to make decisions at the moment in which change results are not expected to occur. The limitations are similar to those in other studies related to management models that recommend further research in this area [48]. Additional predictive methods mentioned in other studies must be considered, for example, supervised learning and unsupervised learning by machines [51].

As a result of this study, the tool composed of predictive methods using RBM can help to address the following knowledge gaps: 1. To understand the methodology and predictive models; 2. To incorporate cancer data statistics to monitor and evaluate the medical treatment of children with cancer; 3. To identify new medical cases that confirm a forecast increase or decrease in certain performance indicators; 4. To verify that data can be converted into useful information for decision-making purposes; 5. To improve policies, programs and public services based on reliable information about the effects of governmental actions on society; 6. To strengthen the empirical practice of RBM in a National Institute of Health’ and 7. To use financial performance indicators to associate the cost factor with medical treatment and with the highest levels of National Institutes of Health complexity.

The knowledge gaps not addressed in this study are as follows: 1. To use nonparametric statistics; 2. To use machine learning statistical methods; 3. To use a larger sample to increase statistical power; 4. To use an automated tool to collect, analyze and present performance indicators results such as Python or R which are programming languages that are versatile tools that go beyond statistical analysis, encompassing web development, automation, and machine learning capabilities. Additionally, they provide robust data visualization options through libraries, enhancing data exploration and presentation. Moreover, their open-source nature ensures widespread accessibility, making them cost-effective alternatives to STAT and 5. To use an artificial intelligence approach.

## 5. Conclusions

Our findings show that the predictive model that best fits real data depends on the process context in which medical services have been provided. LR was tested as a powerful statistical tool that can serve as an effective predictive model that produces meaningful results. The ARIMA model was confirmed as a systematic approach for the predictive modeling of both stationary time series and nonstationary time series data. For nonstationary time series data, the mean and variance are unstable, and they are generally converted to stationary time series by means of a differential operation. Then, stationary time series data are used to establish the ARMA mode [58]. Finally, ES was tested as the easiest forecasting technique and demonstrated that it can be used by businesses of all kinds [17].

The performance of healthcare services is related to their capacity to quickly respond to changes in indicator results; predictive statistical methods with RBM as a framework are fundamental tools to predict future behavior and detect changes as soon as possible to find the reason for variations presented to support the decision-making process to solve problems or improve the quality of medical services by implementing suitable statistical methods and procedures.

Predictive models must be promoted to support RBM. Three predictive models: LR, ES and ARIMA, were compared in this paper: more complex models do not necessarily predict the real values more accurately.

The understanding and use of predictive models in hospital management should be disseminated to improve planning mechanisms and results provided to the served population.

## Figures and Tables

**Figure 1 cancers-15-04649-f001:**
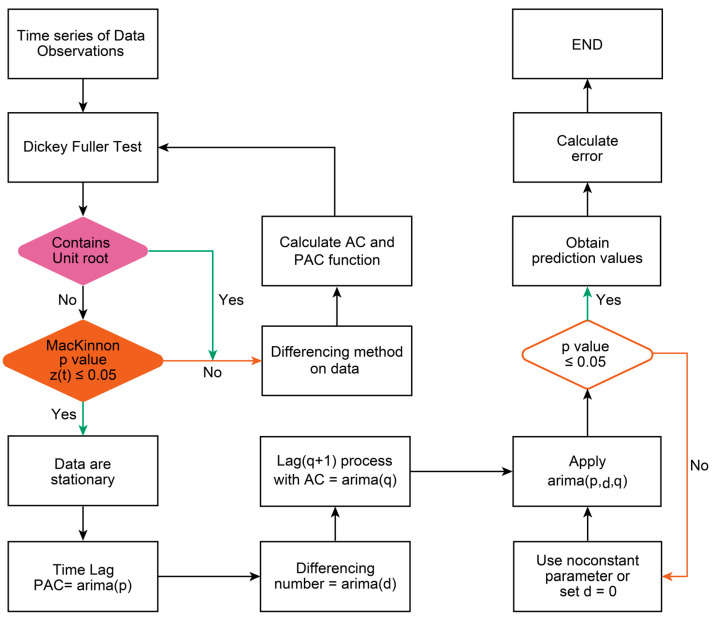
Box–Jenkins-Jenkins/ARIMA process.

**Figure 2 cancers-15-04649-f002:**
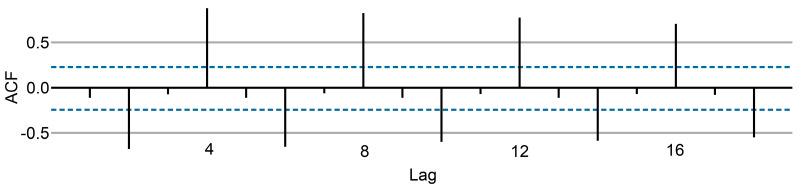
Correlogram of nine values or ACF.

**Figure 3 cancers-15-04649-f003:**
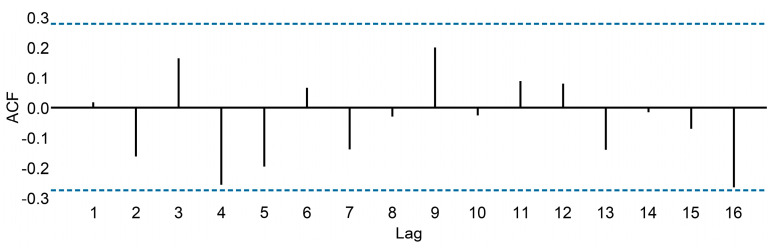
Autocorrelation function for the white noise series.

**Table 1 cancers-15-04649-t001:** Cause of death, cause of morbidity and cause of hospital outpatient for each cancer type of diagnosis.

Year	Cause of Death	Cause of Morbidity	Cause of Hospital Outpatient
2016 3798 patients	Tumors and neoplasm (2)/33.7%/64 patients	Acute Lymphoblastic Leukemia (1)/13.5%/1024 patients Malignant Tumor (3)/2.3%/176 patients	Tumors and neoplasm (1)/33.4%/2534 patients
2017 3768 patients	Tumors and neoplasm (1)/42.7%/73 patients	Acute Lymphoblastic Leukemia (1)/13%/1006 patients Malignant Tumor (2)/2.9%/108 patients	Tumors and neoplasm (1)/33.3%/2581 patients
2018 4130 patients	Tumors and neoplasm (2)/24.3%/43 patients	Acute Lymphoblastic Leukemia (1)/13.3%/1060 patients Malignant Tumor (2)/3.2%/399 patients	Tumors and neoplasm (1)/34%/2628 patients
2019 3591 patients	Tumors and neoplasm (1)/35%/62 patients	Acute Lymphoblastic Leukemia (1)/11.2%/915 patients Malignant Tumor (2)/2.8%/273 patients	Tumors and neoplasm (1)/32.1%/2341 patients
2021 2386 patients	Tumors and neoplasm (1)/34.2%/51 patients	Acute Lymphoblastic Leukemia (1)/12.5%/670 patients Malignant Tumor (2)/1.3%/68 patients	Tumors and neoplasm (1)/29.7%/1597 patients
	Position among the main causes appear in parenthesis/Percentage from total/Number of patients		

**Table 2 cancers-15-04649-t002:** Summary statistics of predictive models.

Performance Indicator’s Name	Predictive Model	MAE	Deviation Standard	*p* Value
Bed occupancy rate	ARIMA(2,0,1)	2.3117	2.2368	0.017 *
	LR	2.2453	1.5129	0.003
	ES	2.5845	1.8873	NA
Hospital admissions through emergency department	ARIMA(3,2,1) noconstant	0.1292	0.0789	0.252 *
	LR	1.1620	1.5129	0.000
	ES	1.1325	1.3363	NA
Bed turnover rate	ARIMA(1,0,1) noconstant	0.5129	0.6295	0.028 *
	LR	0.3320	0.2181	0.004
	ES	0.3841	0.2655	NA
Bed rotation index	ARIMA(1,0,1) noconstant	0.7388	1.6067	0.180 *
	LR	0.3012	0.1925	0.000
	ES	0.3447	0.2225	NA
Percentage of emergency Admissions	ARIMA(1,0,1) noconstant	2.2322	3.3174	0.025 *
	LR	1.3884	1.5952	0.025
	ES	1.4450	1.5600	NA
Major surgery index	ARIMA(1,0,1) noconstant	2.4514	1.8210	−2.756 *
	LR	1.8183	1.5148	0.001
	ES	1.8642	1.8720	NA
Mortality rate	ARIMA(1,0,0) noconstant	5.5113	6.5088	0.805 *
	LR	3.0747	2.2742	0.688
	ES	3.1302	2.2275	NA
Percentage of scheduled consults granted	ARIMA(1,0,0) noconstant	0.9754	2.0528	0.135 *
	LR	0.3594	0.2322	0.151
	ES	0.3887	0.2552	NA
First time consults index	ARIMA(3,0,1) noconstant	2.4413	5.3074	−1.589 *
	LR	1.5087	1.2620	0.483
	ES	1.5472	1.1538	NA
Proportion of subsequent consultants with relation to the first time	ARIMA(4,0,1)	1.5178	1.3511	−2.865 *
	LR	1.3400	1.3552	0.000
	ES	1.7370	1.6261	NA

Abbreviations: LR Linear regression, ES Exponential Smoothing, MAE: mean absolute error; NA: does not apply; *p* < 0.05 there is evidence to reject the null hypothesis. * If no constant parameter is included it is assumed that there is no constant in the trend.

**Table 3 cancers-15-04649-t003:** ARIMA criteria results.

	Criteria ARIMA Used	Value/Type	*p* Value
Bed occupancy rate	dfuller CV z(t) 95% = −3.6	−3.5740	0.0321
	dfuller best difference	0	
	dfuller value, trend, constant (a,b,c)	0.1621 (b)	0.263 (b)
	ARIMA(2,0,1)/IC 95% (ar,ma)	Ma	0.017
Hospital admissions through emergency department	dfuller CV z(t) 95% = −3.6	−5.6590	0.0001
	dfuller best difference	2	
	dfuller value, trend, constant (a,b,c)	0.7877 (c)	0.375 (c)
	ARIMA(1,2,3)/IC 95%	Ma	0.002
Bed turnover rate	dfuller CV z(t) 95% = −3.6	−4.1860	0.0047
	dfuller best difference	0	
	dfuller value, trend, constant (a,b,c)	−0.0350 (b)	0.125 (b)
	ARIMA(1,0,1) noconstant/IC 95% (ar,ma)	Ma	0.028
Beds rotation index	dfuller CV z(t) 95% = −3.6	−4.5910	0.0011
	dfuller best difference	0	
	dfuller value, trend, constant (a,b,c) *	0.0239	0.180 (b)
	ARIMA(1,0,1) noconstant/IC 95% (ar,ma)	Ma	0.000
Percentage of emergency admissions	dfuller CV z(t) 95% = −3.6	−5.4260	0.000
	dfuller best difference	2	
	dfuller value, trend, constant (a,b,c) *	0.1703	0.197 (b)
	ARIMA(4,2,0) noconstant/IC 95% (ar,i)	Ar	0.042 (L2)
Major surgery index	dfuller CV z(t) 95% = −3.6	−4.0760	0.0068
	dfuller best difference	1	
	dfuller value, trend, constant (a,b,c) *	−0.0103	0.942 (b)
	ARIMA(1,1,1) noconstant/IC 95%	Ar	0.973
Mortality rate	dfuller CV z(t) 95% = −3.6	−4.8190	0.0004
	dfuller best difference	0	
	dfuller value, trend, constant (a,b,c) *	0.0405	0.805 (b)
	ARIMA(1,0,0) noconstant/IC 95% (ar)	Ar	0.000
Percentage of scheduled appointments granted	dfuller CV z(t) 95% = −3.6	−3.9990	0.0088
	dfuller best difference	0	
	dfuller value, trend, constant (a,b,c)^*^	0.0405	0.135 (b)
	ARIMA(1,0,0) noconstant/IC 95% (ar)	Ar	0.000
First time consults index	dfuller CV z(t) 95% = −3.6	−3.8820	0.0088
	dfuller best difference	1	
	dfuller value, trend, constant (a,b,c) *	1.1924	0.242 (c)
	ARIMA(2,1,1) noconstant/IC 95%	Ar	0.752 (L1)
Proportion of subsequent consults with relation to the first time	dfuller CV z(t) 95% = −3.6	−3.6460	0.0262
	dfuller best difference	1	
	dfuller value, trend, constant (a,b,c) *	−0.1540	0.902 (c)
	ARIMA(2,1,1) noconstant/IC 95%	Ar	0.646 (L1)

Abbreviations: a: dfuller value; CV: Critical value MacKInnon test; ar: autoregressive; b: trend; c: constant; dfuller: Dickey–Fuller to test unit root in stationary data; L1: first regression; L2: second regression; ma: moving average. * trend constant in the model means that it is assumed that the time series has a constant linear trend over time.

**Table 4 cancers-15-04649-t004:** LR assumptions check (Normal, outlier and heteroscedasticity tests).

Performance Indicator	Shapiro Wilk *p*-Value (1)	Cook’s Distance (2)	Breusch Pagan *p*-Value (3)
Bed occupancy rate	0.027	0 > 1	0.3976
Hospital admissions through emergency department	0.000	2 > 1	0.0001
Bed turnover rate	0.085	0 > 1	0.4146
Bed rotation index	0.409	0 > 1	0.5580
Percentage of emergency admissions	0.000	1 > 1	0.0000
Major surgery index	0.568	0 > 1	0.6507
Mortality rate	0.263	1 > 1	0.4861
Percentage of scheduled consults granted	0.999	0 > 1	0.7536
First time consults index	0.209	0 > 1	0.3049
Proportion of subsequent consultant with relation to the first time	0.899	1 > 1	0.5265

(1) Normality test. (2) The outlier test indicates the number of outliers among all 20 analyzed data values greater than one. (3) Heteroskedasticity test.

## Data Availability

The raw data supporting the conclusions of this article will be made available by the authors without undue reservation.

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
