# Peer review of "Comparative Analysis of Three Predictive Models of Performance Indicators with Results-Based Management: Cancer Data Statistics in a National Institute of Health"

_cancers, 2023, doi:10.3390/cancers15184649_

Round 1

Reviewer 1 Report

This work presents methods of comparison between models predicting decision making in clinical settings.  Particularly, they focus on comparing the performance of simple linear regression, exponential smoothing, and auto-regressive integrated moving average models.  This is a very important work for clinicians with implications in results-based management needed for decision making in diagnosis and treatment.  Therefore, the reviewer recommends this work for publications with minor edits.

1. Lines 157-8.  The authors may want to review specific examples of improper use of linear regression.

2. Lines 209-212.  The authors may want to clarify how use of the statistical methods stated relate to BSC.

3. Line 562, Table 1. Include cause of death and cause of morbility for each type of diagnosis.

Author Response

Comments and Suggestions for Authors

This work presents methods of comparison between models predicting decision making in clinical settings.  Particularly, they focus on comparing the performance of simple linear regression, exponential smoothing, and auto-regressive integrated moving average models.  This is a very important work for clinicians with implications in results-based management needed for decision making in diagnosis and treatment.  Therefore, the reviewer recommends this work for publications with minor edits.

R: Thank you to the reviewer for acknowledging that this is very important work for clinicians and that it has implications in the area of results-based management needed for decision making in diagnosis and treatment. We have thoroughly reviewed each comment and provided appropriate responses.

Therefore, the reviewer recommends this work for publications with minor edits.

  1. Lines 157-8. The authors may want to review specific examples of improper use of linear regression.

R: We have thoroughly reviewed specific examples of improper linear regression use and addressed the wording in the cited lines, providing three clear examples. Additionally, we introduced a validation process to mitigate any further inappropriate application of SLR.

In the Results section, we have included Table 4, which presents the results of the Shapiro‒Wilk test for normal validation, the Breusch‒Pagan test for heteroscedasticity, and Cook's distance to identify influential outliers. These additional analyses strengthen the validity of our findings.

Furthermore, we have made significant modifications to the Results, Discussion, and Abstract sections to emphasize the importance of validating linear regression. By doing so, we have enhanced the credibility and reliability of our research outcomes.

  1. Lines 209-212. The authors may want to clarify how use of the statistical methods stated relate to BSC.

R: We provided clarifications for some of the mentioned statistical methods, specifically the structural equation model (SEM), confirmatory factor analysis (CFA), and the associated fit indices, including the comparative fit index (CFI), Tucker‒Lewis index (TLI), root mean square error of approximation (RMSEA), and standardized root mean square residual (SRMR). Additionally, we emphasize that these statistical methods can be applied to any existing management model, not just the BSC.

  1. Line 562, Table 1. Include cause of death and cause of morbidity for each type of diagnosis.

R: Table 1 has been modified to display the cause of death, cause of morbidity, and cause of hospital outpatient visits for each type of cancer diagnosis. We believe that the revised table is now clearer and aligns with the recommendation.

Reviewer 2 Report

Unfortunately, at the moment I cannot evaluate the scientific value of the manuscript as the paper is difficult to read - it includes many repetitions and many well-known facts (e.g. the description of linear regression is redundant). Also, the English language needs improvement. 

I advise the authors to go carefully through the paper in order to make it mere clear and concise (e.g., the Introduction-section can be reduced to max 2 pages without any harm for understanding). Also, the paper should be revised by a specialist in English language.

The English language needs improvement. The paper should be revised by a specialist in English language.

Author Response

Comments and Suggestions for Authors

Unfortunately, at the moment I cannot evaluate the scientific value of the manuscript as the paper is difficult to read - it includes many repetitions and many well-known facts (e.g. the description of linear regression is redundant).

R: Thank you for the feedback and contributions provided by the reviewer. We have carefully addressed each point, resulting in substantial improvements to the manuscript. In relation to Objective number 2, as mentioned in point "1.4 Problem Statement," we would like to emphasize that we do not consider the description of linear regression to be redundant. Our literature review has revealed that the utilization of RBM models in Mexico is lacking, and monitoring and evaluation are identified as weak aspects of the model. Therefore, our primary aim is to offer a clear and comprehensive description of each statistical method and provide a reference framework to support public organizations in achieving public value (results) and promote its use with cancer statistics data.

Also, the English language needs improvement.

The article underwent evaluation by experts in academic English to ensure compliance with language requirements.

I advise the authors to go carefully through the paper in order to make it mere clear and concise (e.g., the Introduction-section can be reduced to max 2 pages without any harm for understanding). Also, the paper should be revised by a specialist in English language.

R: Thank you for your suggestion. We have taken this into consideration and made the necessary revisions. While we have eliminated some lines and paragraphs in the Introduction section and in point 2.2 Procedure, we believe it is crucial to provide a comprehensive understanding of the RBM model, performance indicators, and the three predictive models for nurses, technicians, medical personnel, and managers. Additionally, we feel it is essential to present other management models before addressing the statement of the problem and the methodology.

As mentioned earlier, the article has undergone evaluation by experts in academic English to meet the requested requirements and ensure its quality.

Reviewer 3 Report

I thank the authors for the important analytic efforts profused and the useful comparison between statistical methods.

The work is for the most part a useful review of statistical analysis techniques for the "Performance Management", in particular the healthcare one. It is written correctly, the analyses are conducted with technical competence. Contact with the Oncological World is present in the date set analysed by the authors but not commented in reference to the specificities to Oncology itself.

Nevertheless, I am very perplexed for two main reasons:

The magazine, according to my opinion, was not chosen correctly. I would suggest a Public Health Journal or in any case referable to the Medical Decision Making area;

The descriptive part of the methods is unduly hypertrophic and requires nothing but specific notes.

Although personally in accordance with the authors when they say "More Complex Models Do Not necessaryly Predict Real Values More Accurately, and the Model That Fits the Data Best Depends 296 On the Medical Process Context in Which the results are obtained" I think the wealth of methods, as indeed implicit in the spirit of the study, contributes to strengthening the conclusions.

In summary, I believe that the work requires a wide work in rewriting methods in a less didactic key and, however, I would suggest a different Journal more linked to health management issues.

Author Response

Comments and Suggestions for Authors

I thank the authors for the important analytic efforts profused and the useful comparison between statistical methods.

R: Thank you for your valuable comment regarding the useful comparison between statistical methods.

The work is for the most part a useful review of statistical analysis techniques for the "Performance Management", in particular the healthcare one. It is written correctly, the analyses are conducted with technical competence. Contact with the Oncological World is present in the date set analysed by the authors but not commented in reference to the specificities to Oncology itself.

R: Thank you  for your comment acknowledging the technical competence of our analyses. Our aim was to provide a detailed description of three statistical methods to illustrate their use in helping administrators, professionals, and decision-makers convert data into valuable information for making informed decisions. We also aimed to identify which method best fits the real expected values, thus providing statistical evidence to analyze cancer data statistics.

The central contribution of our work lies in demonstrating how well the described instrument predicts trends, enabling corrective actions and improving the effectiveness and quality of medical services delivered to children with cancer. The results underscore the importance of implementing performance indicators associated with cancer and analyzing them with predictive models for the monitoring and evaluation of cancer cases.

In the Results section, Table 1 has been modified to illustrate the position of cancer diagnoses among the main causes of death, morbidity, and hospital outpatient visits. Tumors and neoplasms were found to be the main causes of death in 2017, 2019, and 2021, while acute lymphoblastic leukemia was the most common disease in 2016, 2017, 2018, and 2021. Furthermore, tumors and neoplasms were the primary reasons for hospital outpatient visits during the five-year study period.

This study highlights the justification for using predictive models, particularly in the context of oncology's specificities, to enhance decision-making and resource allocation in cancer care.

The above justifies that our work should be published in the prestigious journal "Cancer".

Nevertheless, I am very perplexed for two main reasons:

The magazine, according to my opinion, was not chosen correctly. I would suggest a Public Health Journal or in any case referable to the Medical Decision Making area

R: Thank you for your comment. We firmly believe that the detailed analysis and description of the statistical methods serve as the foundation and justification for their utilization with performance indicators associated with children with cancer. Furthermore, we consider that predictive linear models, such as simple linear regression, ARIMA, and simple exponential smoothing, are essential prerequisites to comprehend the significance of our research that encourages the implementation of predictive models with RBM for monitoring and evaluating cancer data statistics.

Therefore, we believe that the journal has been correctly chosen.

The descriptive part of the methods is unduly hypertrophic and requires nothing but specific notes.

R: Thank you for the recommendation. We have made revisions by removing lines and paragraphs from the Introduction section and point 2.2 Procedure. However, we believe that a detailed description of the three predictive models is necessary to ensure that nurses, technicians, medical professionals, and managers can understand them, aligning with the objectives of our work.

Although personally in accordance with the authors when they say "More Complex Models Do Not necessaryly Predict Real Values More Accurately, and the Model That Fits the Data Best Depends On the Medical Process Context in Which the results are obtained" I think the wealth of methods, as indeed implicit in the spirit of the study, contributes to strengthening the conclusions

R: Table 4 was added to emphasize the significance of the linear regression method. To validate normality, the Shapiro‒Wilk test was employed, while heteroscedasticity was validated using the Breusch‒Pagan test, and influential outliers were identified using Cook’s distance.

We have made modifications to the Results, Discussion, and Abstract sections to reinforce the validation of linear regression and the conclusions. As our work extensively covers the ARIMA model, we believe that providing methodological details of the predictive methods is relevant for a comprehensive understanding of our research.

In summary, I believe that the work requires a wide work in rewriting methods in a less didactic key and, however, I would suggest a different Journal more linked to health management issues.

R: The methods are described in a didactic manner because one of our objectives is to illustrate how predictive models can serve as effective tools for analyzing the results of performance indicators associated with RBM. A well-described and validated model is essential for monitoring and evaluating cancer data statistics.

We appreciate your suggestion that we submit to a different journal. However, we believe that our findings offer strong motivation to support the proposal of using predictive methods with RBM for the analysis of cancer data statistics. These methods are currently not in use at the National Institutes of Health, where the research was conducted. Therefore, this study underscores the importance of designing and implementing such performance indicators to benefit children with cancer.

Round 2

Author Response

Thank you again for the opportunity to resubmit our manuscript. I appreciate any assistance that you can provide.

** 

Comments and Suggestions for Authors

The manuscript is still difficult to read as the overburden description of many existing (but not used in the study) predictive models steals the focus from the main purpose of the study - which is to compare three specific models. I suggest moving the overview and description of predictive models from the Introduction and Methods sections to Supplementary Material.

R: We sincerely appreciate the valuable feedback and contributions provided by the reviewer. We have thoroughly addressed each point you raised, and your insights have significantly enhanced the manuscript. Regarding the suggestion to relocate the overview and description of predictive models from the Introduction and Methods section to the Supplementary Material, we believe that implementing this change may not be advisable as it could be perceived as disregarding the input of the other two reviewers. Notably, reviewer 3 has confirmes that the authors provided a good revision and the study deserves publication, and reviewer 1 offered no comments during the second review round, while during the initial review, he requested only minor edits.

In our response to reviewer 3 during round 1, we emphasized the importance of providing a comprehensive description of the three predictive models. This ensures that nurses, technicians, medical professionals, and managers can comprehend the models, which aligns with the objectives of our study. We also highlighted our belief that a thorough analysis and explanation of the predictive models are essential for their proper utilization, particularly in relation to performance indicators for children with cancer. Anyway, the content was reviewed by eliminating certain lines and paragraphs from the Introduction and Methods sections to enhance clarity and eliminate redundancy.

At the same time the important details regarding the study are missing, e.g. sample size. See my specific points beneath.

Summary:

  1. line 17: the word” worldwide” is not necessary and can be avoided.

R: The word “worldwide” was eliminated.

  1. line 25: the word “must” is very strong for this context and should be replaced, for example by “can”.

R: The word “must” was modified an now appears as “can”.

Abstract:

  1. line 33: Drop “(ES)” (as it is used only once in the Abstract)

R: The “(ES)” was droped in the abstract.

  1. line 34: Drop “(ARIMA)” (as it is used only once in the Abstract)

R: The “(ARIMA)” was droped in the abstract.

  1. line 35: Drop “(MAE)” (as it is used only once in the Abstract)

R: The “(MAE)” was droped in the abstract.

  1. line 36: These tests were not used to validate SLR, but to check the SLR’s assumptions.

R: The “SLR” was modifed and now appears as “Linear regression” in the abstract and its redaction changes to “Three of their assumptions were checked” and is highlighted in yellow color.

  1. lines 40: Change “number” to “numbers” (it should be plural as you refer to several entities (death,morbidity, outpatient) and use verbum “were” on line 41)

R: The word “number” was modifed and now appears as “numbers” in the abstract, and we made sure to use “were” verbum and are highlighted in yellow color.

Sections 1.2 – 1.3 and 2.3:

  1. Move these sections to Supplementary Material.
  2. Please refer to the initial response in this document, which confirms that “we believe that implementing this change may not be advisable as it could be perceived as disregarding the input of the other two reviewers. Notably, reviewer 3 has confirmes that the authors provided a good revision and the study deserves publication and reviewer 1 offered no comments during the second review round, while during the initial review, he requested only minor edits”; additionally, please review the second paragraph to comprehend our reasons for retaining these sections and our affirmation that the wording has been reviewed and it has been reduced for accuracy.
  3. Check that abbreviations are used appropriately (e.g., don’t use abbreviation if a term is used only once; introduce an abbreviation the first time a term is mentioned; don’t use the full wording for a term after its abbreviation has been introduced).
  4. The abbreviations were reviewed to ensure their appropriate usage in relation to ARIMA, Linear Regression (LR), Exponential Smoothing (ES), and mean absolute error (MAE). These abbreviations are introduced the first time each term is mentioned and are highlighted in yellow at the rest of the manuscript.
  5. In the overview of the models, I would not recommend mentioning the tests you used for checking the LR assumptions (Shapiro-Wilk, etc.) as they are not necessarily the best choices. The tests based on p-values often flag unimportant differences because they do qualify as significant for large sample sizes, and the opposite problem exists for small samples. For example, the best normality test would be a normal probability plot, i.e., a quantile-quantile plot of observed values vs. normal quantiles. This plot will tell you exactly how your data differ from a normal.
  6. The Shapiro-Wilk test emerges as a competitive contender in a study that compare the power of 40 normality tests in which the goodness-of-fit hypothesis was tested for 15 data distributions with 6 different sample sizes. Arnastauskaite, J.; Ruzgas, T.; Braženas, M. An Exhaustive Power Comparison of Normality Tests.Mathematics 2021, 9, 788. https://doi.org/10.3390/math9070788.

The Cook's distance test is present in other studies as significantly greater than an alternative model, the test exhibited a serious robustness issue in outlier scenarios, such as the COVID-19 pandemic. Hung, M.-C.; Ching, Y.-K.; Lin, S.-K. Impact of COVID-19 on the Robustness of the Probability of Default Estimation Model. Mathematics 2021, 9, 3087. https://doi.org/10.3390/math9233087

The Breush–Pagan test for heteroskedasticity check is used in a study that use regression analysis and showed the adequacy of the model’s results. Wołowiec, T.; Kolosok, S.; Vasylieva, T.; Artyukhov, A.; Skowron, Ł.; Dluhopolskyi, O.; Sergiienko, L. Sustainable Governance, Energy Security, and Energy Losses of Europe in Turbulent Times. Energies 2022, 15, 8857. https://doi.org/10.3390/en15238857

The Q-Q plots exported from Stata in TIFF format are presented for the seven indicators with p-values < 0.05 (Table 2 in the manuscript). These plots exhibit conformity to the assumptions made for the Shapiro-Wilk, Cook's distance, and Breusch-Pagan tests (Table 4). It is evident that plots 3 and 7 exhibit deviations from the normality assumptions, consistent with the test results. Therefore, our selections prove to be valuable for the study's objectives.

1. Bed turnover rate

2. Bed rotation index

3. Percentage of emergency admissions

4. Major surgery index

5. Proportion of subsequent consultant with relation to the first time

6. Bed occupation rate

7. Hospital admissions through emergency department

Finally, it's of importance to emphasize that the assessment of linearity assumptions was incorporated due to the invaluable input from Reviewer 1 during the initial review round. Moreover, your insightful recommendations and observations regarding the errors identified in your comprehensive review have contributed to strengthening the manuscript.

We have conscientiously addressed all the concerns you raised, thoroughly reviewed the literature, and presented the Q-Q plot. However, we have chosen not to include these in the final manuscript, as our aim is to showcase the utility of the assumption tests employed for the scope of our study. This approach has significantly elevated the overall quality of the document, thanks to your contribution.

  1. Another problem with Shapiro-Wilk etc. tests is your wrong interpretation of the test’s results. For example, you write (line 144) that you test the null hypothesis that the distribution of the residuals is normal, and if p<0.05 then the null hypothesis is rejected and you conclude that the data is normally distributed. However, this is wrong, as if p<0.05 than conclusion is that the data is not normally distributed. Contact a statistician for an explanation if necessary.
  2. We greatly appreciate the reviewer's assistance in identifying the incorrect interpretation of the test results, which have been rectified, as well as the observations regarding the null hypotheses' association with p-values and conclusions about the linearity of the analyzed data. The document now states as follow:

Point 1.2.4 Linear regression model:

To test the null hypothesis that a given dataset follows a normal distribution, the Shapiro‒Wilk test could be used, if the p-value > 0.05 then we fail to reject the null hypothesis, suggesting that the data is normally distributed[23]. 2. LR should check homoskedasticity assumption, which is a desirable property where the variability of the errors (residuals) is constant across all levels of the independent variable. One way to detect heteroskedasticity is by using the Breusch‒Pagan test, the null hypothesis is that there is homoskedasticity in a LR model. If the p-value > 0.05 we fail to reject the null hypothesis, suggesting homoskedasticity, the problem with heteroskedasticity is that it may lead to incorrect estimates of the standard errors for the coefficients and, consequently get incorrect t values [24].

One way to detect heteroskedasticity is by using the Breusch‒Pagan test, the null hypothesis is that there is homoskedasticity in a LR model. If the p-value > 0.05 we fail to reject the null hypothesis, suggesting homoskedasticity,

Comments at the end of table 4 in the results section now indicate:

Seven performance indicators show normality according with saphiro wilk test, the first two, belong to the seven indicators with better adjusted results and must be interpreted with caution, the remaining four, fulfill the requirement of displaying a linear pattern.

Finally, the breush pagan test display that nine out of the ten indicators show homoskedasticity; therefore, the coefficient errors are being estimated correctly

In the Discussion section, the limitations were documented as follows:

As noted, limitations of the study are the lack of probabilistic sampling; using p-values for decision-making should be made with caution due to dependence on sample-size: for large samples p-values can be small even if difference doesn’t exist and the other way around; small variations in data or analysis methods can lead to significant changes in p-values, affecting the interpretation of results; the tests used for checking the LR assumptions (Shapiro-Wilk, Cook’s distance and Breusch pagan tests) are not necessarily the best choices because the tests based on p-values often flag unimportant differences because they do qualify as significant for large sample

sizes, and the opposite problem exists for small samples; exists more statistical tests that were not used to check LR assumptions as: independence of errors, absence of multicollinearity, no autocorrelation and constant variance.

  1. You use several names for linear regression (linear regression, simple linear regression, single linear regression). Please use only one name throughout the manuscript.

R: We opted to utilize "Linear regression (LR)" and thoroughly reviewed the manuscript to ensure consistent usage its abbreviation.

  1. You use several names for exponential smoothing (exponential smoothing, simple exponential smoothing). Please use only one name throughout the manuscript.

R: We opted to utilize "Linear regression (LR)" and thoroughly reviewed the manuscript to ensure consistent usage its abbreviation.

  1. Show us an example of your models in a Supplementary material. For example, write a linear regression equation for one of ten performance indicators

R: The linear regression equiation for “Bed occupancy rate” is as follow:

Y = 83.047 + 0.381 x

Since no supplementary material was added due to the aforementioned reasons, the equation is solely presented in this document as a response.

  1. Lines 277-282: Drop this paragraph as it mostly repeats the previous paragraph (lines 271-276).

R: The paragraph was droped.

  1. Lines 284-285: Drop as MAE is described further in the Methods.

R: The lines was droped.

Design:

  1. How many children were included in the study (for each year)?

R: Table 1 displays the count of patients, all of whom are children and adolescents. The corresponding details are presented in the manuscript as follows

In the point 2.1 Design, was documented as follows:

“number of patients and cancer diagnostics were included (Table 1)”

Table 1 changed as follows:

Table 1. Cause of death, cause of morbidity and cause of hospital outpatient for each cancer type of diagnosis.

Year

Cause of death

Cause of morbidity

Cause of Hospital Outpatient

2016

3,798 patients

Tumors and neoplasm (2)/33.7%/64 patients

Acute Lymphoblastic Leukemia (1)/13.5%/1,024 patients
Malignant Tumor (3)/2.3%/176 patients

Tumors and neoplasm (1)/33.4%/2,534 patients

2017

3,768 patients

Tumors and neoplasm (1)/42.7%/73 patients

Acute Lymphoblastic Leukemia

(1)/13%/1,006 patients
Malignant Tumor (2)/2.9%/108 patients

Tumors and neoplasm (1)/33.3%/2,581 patients

2018

4,130 patients

Tumors and neoplasm (2)/24.3%/43 patients

Acute Lymphoblastic Leukemia

(1)/13.3%/1,060 patients
Malignant Tumor (2)/3.2%/399 patients

Tumors and neoplasm (1)/34%/2,628 patients

2019

3,591 patients

Tumors and neoplasm (1)/35%/62 patients

Acute Lymphoblastic Leukemia

(1)/11.2%/915 patients
Malignant Tumor (2)/2.8%/273 patients

Tumors and neoplasm (1)/32.1%/2,341 patients

2021

2,386 patients

Tumors and neoplasm (1)/34.2%/51 patients

Acute Lymphoblastic Leukemia

(1)/12.5%/670 patients
Malignant Tumor (2)/1.3%/68 patients

Tumors and neoplasm (1)/29.7%/1,597 patients

Position among the main causes appear in parenthesis/Percentage from total/Number of patients

  1. Specify please at which one of the National Institutes the data were collected.

R: The revised Hospital’s name in “Section 2.1 Design” is now presented as follows:

“ …10 medical performance indicator results to evaluate the performance of the Hospital Infantil de Mexico Federico Gómez which is one of the National Institutes of Health in Mexico”.

  1. Which specialists collected and analysed the data?

R: “Data were collected and analysed by the authors of this paper..”

  1. Line 293: Drop “and existence of results in the analyzed period” (as it is obvious from line 292).

R: The line was droped.

  1. Line 296: Change “3 models” to “3 models (ES, ARIMA and LR)”

R: Line change to: 3 models were evaluated (ARIMA, LR and ES).

  1. Lines 296-297: These two sentences belong rather to Procedure-section. (One of them is already there – see line 304).

R: The two sentences were moved to procedure section.

  1. Lines 298-299: This sentence belongs rather to Statistical Analysis section

R: The sentences were moved to statistical analysis section.

Procedure:

  1. Line 300: Procedure for what?

R: The line change to say “Procedure for measuring the performance of each model”.

  1. The sentence starting with “We made ..” can be shortened to “We made quarterly forecasts of performance indicator results with four quarterly lagged results as a predictor”.

R: The sentence change to say “We made quarterly forecasts of performance indicator results with four quarterly lagged results as a predictor” as the reviewer recommended.

  1. Line 307: Drop “of the 3 statistical predictive models” (as it is obvious from the previous text).

R: The line was droped.

Statistical Analysis:

  1. You don’t need to mention so many times that significance level chosen was 5%. Just mention it once.

R: The sentence change to say “We made quarterly forecasts of performance indicator results with four quarterly lagged results as a predictor” as the reviewer recommended.

  1. Which statistical tests were used to derive p-values?

R: The sentences were modified to provide specific details for ARIMA, LR, and ES, aand now is indicated in the manuscript as follow.

A value of p < 0.05 is used to provides evidence to reject the null hypothesis in favor of the alternative hypothesis. The ARIMA Dickey-Fuller test was employed with the null hypothesis that time series data has a unit root equal to 1 or it is non-stationary. The alternative hypothesis asserts that the root differs from 1, signifying the absence of a unit root and ensuring that the data is stationary with a stable mean and variance. Addition-ally, the MacKinnon test was utilized for its enhanced precision in critical values, partic-ularly for small sample sizes. This test represents an improved iteration of the Dick-ey-Fuller test and aligns with the aforementioned logic.

For LR model the null hypothesis is that the coefficient of the independent variable (time) is equal to zero, meaning that do not have a significant effect on the dependent var-iable (performance indicator results). The alternative hypothesis, in contrast, asserts that there is a significant linear relationship between the variables. A significant p-value in linear regression means that there is sufficient evidence to reject the null hypothesis and suggests that there is a relationship between the independent variable and the dependent variable.

For ES model, the “desired responsiveness index or optimal factor α” was calculated to obtain the predicted values. ES is not directly associated with a p-value as in statistical hypothesis testing. Instead, exponential smoothing is a forecasting method that uses a weighted averaging approach to predict future values in a time series.

  1. Lines 472-474: Drop as this information has already been given earlier.

R: The lines were droped.

  1. Line 474: Change “number” to “numbers”

R: The word “number” cahnged to “numbers”.

  1. Lines 479-481: Drop as it has been already explained in the Procedure-section.

R: The lines were droped.

  1. Line 482: Not clear, what results.

R: The redaction was changed to:

For the first specific objective, the MAE result for each model was used to identify the predictive model with the smallest error between the real and predictive values.

  1. Line 483: Not clear, variation of what.

R: The redaction was changed to:

The smallest standard deviation also was considered as an indication of data consistency and homogeneity, as it resulted in better predictive values in relation to the mean.

  1. Line 482, regarding p-value: What was the null hypothesis?

R: Please review the response provided above in number 28.

  1. Line 488: Change “The Dockey-Fuller test” to “For ARIMA method, the Dockey-Fuller test“.

R: The modification was implemented as follows:

“For ARIMA method, the Dickey-Fuller test“

  1. Line 487: Drop (see comment 35)

R: The line was droped.

  1. Line 508, regarding p-value: What was the null hypothesis?

R: Please review the response provided above in number 28.

  1. Line 515, regarding p-value: What was the null hypothesis?

R: Sentences changed to:

For the third specific objective to prove the validity of the RBM in relation to the per-formance indicator results analysis with predictive models, the best model can be used in order to calculate future expected values.

  1. Line 516: standard deviation for what?

R: Same as in 38; Sentences changed to:

For the third specific objective to prove the validity of the RBM in relation to the per-formance indicator results analysis with predictive models, the best model can be used in order to calculate future expected values.

Results:

  1. Table 1: Add total numbers.

R: Table 1 now shows the total number of patients, see response to number 17.

  1. Table 2: Move performance indicators titles to the left, otherwise it is difficult to read the table.

R: Table 2 has been enhanced to incorporate performance indicator titles within an extra left-hand column; similarly, table 3 has also been adjusted to encompass the same suggested alteration.

  1. Table 2: “*” is used in column “P Value” without explanation in the legend.

R: At the end of table 2, an explanatory note regarding the p-value has been appended, stating as follow:

Abbreviations: LR Linear regression, ES Exponential Smoothing, MAE: mean absolute error; NA: does not apply; p < 0.05 there is evidence to reject the null hypothesis.

  1. Line 533: Wrong numbering. Change the section number to 3.3 (with corresponding change in numbering of further sections)

R: The section numbers have been rectified to 3.3, 3.3.1, 3.3.2, and 3.3.3.

  1. Table 3, legend: what is “movil”?

R: The word movil have been rectified to moving.

  1. Line 557: Refer to a table.

R: Table 2 has been added and enclosed within parentheses as a reference.

  1. Line 558: Table 3 doesn’t include any results for linear regression.

R: Table 3 has been updated to Table 2.

  1. Line 560 and Table 4: “validation” is a wrong term here. Using these methods, you don’t validate LR, but check its assumptions.

R: The terms were altered as follows:

“The assumptions were checked as shown in Table 4.”

  1. Table 4: Very confusing. If it is about LR (as the title states), why other methods are mentioned in column 1?

R: The methods listed in column 1 have been removed.

  1. Table 4: Not clear, what mathematical terms the numbers in columns 2-4 represent. If p-values, then its interpretation in lines 567-573 is wrong (see comment 11).

R: The interpretation has been rectified as outlined below:

Seven performance indicators demonstrate normality as per the Shapiro-Wilk test. However, the first two indicators and the percentage of emergency admissions do not exhibit a normal distribution. It's important to note that since p > 0.05, we do not reject the null hypothesis suggesting that the data follows a normal distribution. We have effectively addressed outliers; within the 20 data points analyzed, three indica-tors have one or two data points exceeding the threshold. This indicates that the simple linear regression model is not significantly affected by unusual values. Lastly, the Breusch-Pagan test reveals that nine out of the ten indicators show homoskedasticity, confirming the accurate estimation of coefficient errors. Similarly, as with the Shapiro-Wilk test, the p-value > 0.05 supports that null hypothesis coul not be rejected suggesting homoskedasticity.

  1. Section 3.2.3: Refer to a table.

R: Table 2 has been included and placed within parentheses as a reference. Section 3.2.3 has been corrected to 3.3.3 (refer to number 43).

Discussion:

  1. Line 614: What is NIH?

R: was modified to: “at the Hospital Infantil de Mexico Federico Gomez”.

  1. Lines 615-819: Statements regarding validation, accuracy and reliability are wrong. The tests mentioned were used only to check the methods assumptions.

R: was modified to:

To check methods assumptions of the LR model, we employed the Shapiro-Wilk, Cook's distance, and Breusch-Pagan tests. Our results demonstrate that the model fits appropriately for the majority of the performance indicators.

  1. Limitations: Discuss limitations regarding using p-values for decision-making. E.g., dependence of p values on sample-size: for large samples p-values can be small even if difference doesn’t exist and the other way around.

R: was modified to:

It's worth noting that the study has several limitations. Firstly, the absence of probabilistic sampling impacts the generalizability of findings. Secondly, the utilization of p-values for decision-making warrants caution due to their sensitivity to sample size; in large samples, p-values may appear small even when differences are absent, and vice versa. Additionally, minor fluctuations in data or analytical methodologies can lead to substantial variations in p-values, influencing result interpretations.

  1. Limitations: Discuss limitations of statistical tests you used to check assumptions of LR.

R: was modified to:

Moreover, the tests chosen to assess LR assumptions (Shapiro-Wilk, Cook's dis-tance, and Breusch-Pagan tests) might not be the most suitable options and could ben-efit from the inclusion of Q-Q plots. Tests relying on p-values can sometimes detect in-significant differences as significant in larger sample sizes, while the reverse problem arises for smaller samples. Notably, there are other statistical tests available that were not utilized to assess LR assumptions, such as assessing independence of errors, de-tecting multicollinearity, identifying autocorrelation, and verifying constant variance.

  1. Line 679: What do you mean under “automated tool”? (Is Stata not “automated” enough compared to Phyton and R?

R: was modified to:

To use an automated tool to collect, analyze and present performance indicator results such as Python or R which are programming languages are versatile tools that go beyond statistical analysis, encompassing web development, automation, and machine learning capabilities. Additionally, they provide robust data visualization options through librar-ies, enhancing data exploration and presentation. Moreover, their open-source nature en-sures widespread accessibility, making them cost-effective alternatives to STAT.

Conclusions

We greatly value your feedback and the errors you've pointed out, our aim was to provide a detailed description of three statistical methods to illustrate their use in helping administrators, professionals, and decision-makers convert data into valuable information for making informed decisions. We also aimed to identify which method best fits the real expected values, thus providing statistical evidence to analyze cancer data statistics.

The central contribution of our work lies in demonstrating how well the described instrument predicts trends, enabling corrective actions and improving the effectiveness and quality of medical services delivered to children with cancer. The results underscore the importance of implementing performance indicators associated with cancer and analyzing them with predictive models for the monitoring and evaluation of cancer cases.

The modifications carried out have reinforced the key points that underscore the significance of this study, serving as a rationale for the utilization of predictive models, especially within the unique landscape of oncology. These adjustments further emphasize the role of predictive models in improving decision-making and optimizing resource allocation in the realm of cancer care.

The above justifies that our work deserves to be published in the prestigious journal "Cancer".

Reviewer 3 Report

I think the authors provided a good revision and the study deserves publication

Author Response

 August 29, 2023, Mexico City

 Responses to reviewers

Comparative Analysis of Three Predictive Models of Performance Indicators with Results-Based Management: Cancer Data Statistics in a National Institute of Health 2506579.

Suhaylah Ingar
Assistant Editor MDPI

Cancers

Dear Editor:

Thank you for the opportunity to resubmit our manuscript (Manuscript ID: Cancers- 2506579) titled “Comparative Analysis of Three Predictive Models of Performance Indicators with Results-Based Management: Cancer Data Statistics in a National Institute of Health”. We appreciate the opportunity to publish in Cancers. We carefully reviewed the reviewers’ valuable comments. Because we wish to ensure that we have fully addressed all concerns, we have revised our manuscript based on the reviewers’ suggestions.

Below, we include our point-by-point responses regarding the reviewers’ feedback and describe in detail the changes made to the manuscript. All changes in the manuscript are highlighted in yellow.

Thank you again for the opportunity to resubmit our manuscript. I appreciate any assistance that you can provide.

Sincerely,

Filiberto Toledano-Tolkedano, Ph.D.

Hospital Infantil de México Federico Gómez National Institute of Health.

Márquez 162, Doctores, Cuauhtémoc, México City, 06720, México.

+ 52 55 52289917, ext. 4318. E-mail: [email protected]

Reviewer 1

No comments are available for round 2.

Reviewer 2

Comments and Suggestions for Authors

The manuscript is still difficult to read as the overburden description of many existing (but not used in the study) predictive models steals the focus from the main purpose of the study - which is to compare three specific models. I suggest moving the overview and description of predictive models from the Introduction and Methods sections to Supplementary Material.

R: We sincerely appreciate the valuable feedback and contributions provided by the reviewer. We have thoroughly addressed each point you raised, and your insights have significantly enhanced the manuscript. Regarding the suggestion to relocate the overview and description of predictive models from the Introduction and Methods section to the Supplementary Material, we believe that implementing this change may not be advisable as it could be perceived as disregarding the input of the other two reviewers. Notably, reviewer 3 has confirmes that the authors provided a good revision and the study deserves publication, and reviewer 1 offered no comments during the second review round, while during the initial review, he requested only minor edits.

In our response to reviewer 3 during round 1, we emphasized the importance of providing a comprehensive description of the three predictive models. This ensures that nurses, technicians, medical professionals, and managers can comprehend the models, which aligns with the objectives of our study. We also highlighted our belief that a thorough analysis and explanation of the predictive models are essential for their proper utilization, particularly in relation to performance indicators for children with cancer. Anyway, the content was reviewed by eliminating certain lines and paragraphs from the Introduction and Methods sections to enhance clarity and eliminate redundancy.

At the same time the important details regarding the study are missing, e.g. sample size. See my specific points beneath.

Summary:

  1. line 17: the word” worldwide” is not necessary and can be avoided.

R: The word “worldwide” was eliminated.

  1. line 25: the word “must” is very strong for this context and should be replaced, for example by “can”.

R: The word “must” was modified an now appears as “can”.

Abstract:

  1. line 33: Drop “(ES)” (as it is used only once in the Abstract)

R: The “(ES)” was droped in the abstract.

  1. line 34: Drop “(ARIMA)” (as it is used only once in the Abstract)

R: The “(ARIMA)” was droped in the abstract.

  1. line 35: Drop “(MAE)” (as it is used only once in the Abstract)

R: The “(MAE)” was droped in the abstract.

  1. line 36: These tests were not used to validate SLR, but to check the SLR’s assumptions.

R: The “SLR” was modifed and now appears as “Linear regression” in the abstract and its redaction changes to “Three of their assumptions were checked” and is highlighted in yellow color.

  1. lines 40: Change “number” to “numbers” (it should be plural as you refer to several entities (death,morbidity, outpatient) and use verbum “were” on line 41)

R: The word “number” was modifed and now appears as “numbers” in the abstract, and we made sure to use “were” verbum and are highlighted in yellow color.

Sections 1.2 – 1.3 and 2.3:

  1. Move these sections to Supplementary Material.
  2. Please refer to the initial response in this document, which confirms that “we believe that implementing this change may not be advisable as it could be perceived as disregarding the input of the other two reviewers. Notably, reviewer 3 has confirmes that the authors provided a good revision and the study deserves publication and reviewer 1 offered no comments during the second review round, while during the initial review, he requested only minor edits”; additionally, please review the second paragraph to comprehend our reasons for retaining these sections and our affirmation that the wording has been reviewed and it has been reduced for accuracy.
  3. Check that abbreviations are used appropriately (e.g., don’t use abbreviation if a term is used only once; introduce an abbreviation the first time a term is mentioned; don’t use the full wording for a term after its abbreviation has been introduced).
  4. The abbreviations were reviewed to ensure their appropriate usage in relation to ARIMA, Linear Regression (LR), Exponential Smoothing (ES), and mean absolute error (MAE). These abbreviations are introduced the first time each term is mentioned and are highlighted in yellow at the rest of the manuscript.
  5. In the overview of the models, I would not recommend mentioning the tests you used for checking the LR assumptions (Shapiro-Wilk, etc.) as they are not necessarily the best choices. The tests based on p-values often flag unimportant differences because they do qualify as significant for large sample sizes, and the opposite problem exists for small samples. For example, the best normality test would be a normal probability plot, i.e., a quantile-quantile plot of observed values vs. normal quantiles. This plot will tell you exactly how your data differ from a normal.
  6. The Shapiro-Wilk test emerges as a competitive contender in a study that compare the power of 40 normality tests in which the goodness-of-fit hypothesis was tested for 15 data distributions with 6 different sample sizes. Arnastauskaite, J.; Ruzgas, T.; Braženas, M. An Exhaustive Power Comparison of Normality Tests.Mathematics 2021, 9, 788. https://doi.org/10.3390/math9070788.

The Cook's distance test is present in other studies as significantly greater than an alternative model, the test exhibited a serious robustness issue in outlier scenarios, such as the COVID-19 pandemic. Hung, M.-C.; Ching, Y.-K.; Lin, S.-K. Impact of COVID-19 on the Robustness of the Probability of Default Estimation Model. Mathematics 2021, 9, 3087. https://doi.org/10.3390/math9233087

The Breush–Pagan test for heteroskedasticity check is used in a  study that use regression analysis and showed the adequacy of the model’s results. Wołowiec, T.; Kolosok, S.; Vasylieva, T.; Artyukhov, A.; Skowron, Ł.; Dluhopolskyi, O.; Sergiienko, L. Sustainable Governance, Energy Security, and Energy Losses of Europe in Turbulent Times. Energies 2022, 15, 8857. https://doi.org/10.3390/en15238857

The Q-Q plots exported from Stata in TIFF format are presented for the seven indicators with p-values < 0.05 (Table 2 in the manuscript). These plots exhibit conformity to the assumptions made for the Shapiro-Wilk, Cook's distance, and Breusch-Pagan tests (Table 4). It is evident that plots 3 and 7 exhibit deviations from the normality assumptions, consistent with the test results. Therefore, our selections prove to be valuable for the study's objectives.

1.     Bed turnover rate

2.     Bed rotation index

3.               Percentage of emergency admissions

4.     Major surgery index

5.      Proportion of subsequent consultant with relation to the first time

6.     Bed occupation rate

7.     Hospital admissions through emergency department

Finally, it's of importance to emphasize that the assessment of linearity assumptions was incorporated due to the invaluable input from Reviewer 1 during the initial review round. Moreover, your insightful recommendations and observations regarding the errors identified in your comprehensive review have contributed to strengthening the manuscript.

We have conscientiously addressed all the concerns you raised, thoroughly reviewed the literature, and presented the Q-Q plot. However, we have chosen not to include these in the final manuscript, as our aim is to showcase the utility of the assumption tests employed for the scope of our study. This approach has significantly elevated the overall quality of the document, thanks to your contribution.

  1. Another problem with Shapiro-Wilk etc. tests is your wrong interpretation of the test’s results. For example, you write (line 144) that you test the null hypothesis that the distribution of the residuals is normal, and if p<0.05 then the null hypothesis is rejected and you conclude that the data is normally distributed. However, this is wrong, as if p<0.05 than conclusion is that the data is not normally distributed. Contact a statistician for an explanation if necessary.
  2. We greatly appreciate the reviewer's assistance in identifying the incorrect interpretation of the test results, which have been rectified, as well as the observations regarding the null hypotheses' association with p-values and conclusions about the linearity of the analyzed data. The document now states as follow:

Point 1.2.4 Linear regression model:

To test the null hypothesis that a given dataset follows a normal distribution, the Shapiro‒Wilk test could be used, if the p-value > 0.05 then we fail to reject the null hypothesis, suggesting that the data is normally distributed[23]. 2. LR should check homoskedasticity assumption, which is a desirable property where the variability of the errors (residuals) is constant across all levels of the independent variable. One way to detect heteroskedasticity is by using the Breusch‒Pagan test, the null hypothesis is that there is homoskedasticity in a LR model. If the p-value > 0.05 we fail to reject the null hypothesis, suggesting homoskedasticity, the problem with heteroskedasticity is that it may lead to incorrect estimates of the standard errors for the coefficients and, consequently get incorrect t values [24]. 

One way to detect heteroskedasticity is by using the Breusch‒Pagan test, the null hypothesis is that there is homoskedasticity in a LR model. If the p-value > 0.05 we fail to reject the null hypothesis, suggesting homoskedasticity,

Comments at the end of table 4 in the results section now indicate:

Seven performance indicators show normality according with saphiro wilk test, the first two, belong to the seven indicators with better adjusted results and must be interpreted with caution, the remaining four, fulfill the requirement of displaying a linear pattern.

Finally, the breush pagan test display that nine out of the ten indicators show homoskedasticity; therefore, the coefficient errors are being estimated correctly

In the Discussion section, the limitations were documented as follows:

As noted, limitations of the study are the lack of probabilistic sampling; using p-values for decision-making should be made with caution due to dependence on sample-size: for large samples p-values can be small even if difference doesn’t exist and the other way around; small variations in data or analysis methods can lead to significant changes in p-values, affecting the interpretation of results; the tests used for checking the LR assumptions (Shapiro-Wilk, Cook’s distance and Breusch pagan tests) are not necessarily the best choices because the tests based on p-values often flag unimportant differences because they do qualify as significant for large sample

sizes, and the opposite problem exists for small samples; exists more statistical tests that were not used to check LR assumptions as: independence of errors, absence of multicollinearity, no autocorrelation and constant variance.

  1. You use several names for linear regression (linear regression, simple linear regression, single linear regression). Please use only one name throughout the manuscript.

R: We opted to utilize "Linear regression (LR)" and thoroughly reviewed the manuscript to ensure consistent usage its abbreviation.

  1. You use several names for exponential smoothing (exponential smoothing, simple exponential smoothing). Please use only one name throughout the manuscript.

R: We opted to utilize "Linear regression (LR)" and thoroughly reviewed the manuscript to ensure consistent usage its abbreviation.

  1. Show us an example of your models in a Supplementary material. For example, write a linear regression equation for one of ten performance indicators

R: The linear regression equiation for “Bed occupancy rate” is as follow:

Y = 83.047 + 0.381 x

Since no supplementary material was added due to the aforementioned reasons, the equation is solely presented in this document as a response.

  1. Lines 277-282: Drop this paragraph as it mostly repeats the previous paragraph (lines 271-276).

R: The paragraph was droped.

  1. Lines 284-285: Drop as MAE is described further in the Methods.

R: The lines was droped.

Design:

  1. How many children were included in the study (for each year)?

R: Table 1 displays the count of patients, all of whom are children and adolescents. The corresponding details are presented in the manuscript as follows

In the point 2.1 Design, was documented as follows:

“number of patients and cancer diagnostics were included (Table 1)”

Table 1 changed as follows:

Table 1. Cause of death, cause of morbidity and cause of hospital outpatient for each cancer type of diagnosis.

Year

Cause of death

Cause of morbidity

 Cause of Hospital Outpatient

2016

3,798 patients

Tumors and neoplasm (2)/33.7%/64 patients

Acute Lymphoblastic Leukemia (1)/13.5%/1,024 patients
Malignant Tumor (3)/2.3%/176 patients

Tumors and neoplasm (1)/33.4%/2,534 patients

2017

3,768 patients

Tumors and neoplasm (1)/42.7%/73 patients

Acute Lymphoblastic Leukemia

(1)/13%/1,006 patients
Malignant Tumor (2)/2.9%/108 patients

Tumors and neoplasm (1)/33.3%/2,581 patients

2018

4,130 patients

Tumors and neoplasm (2)/24.3%/43 patients

Acute Lymphoblastic Leukemia

(1)/13.3%/1,060 patients
Malignant Tumor (2)/3.2%/399 patients

Tumors and neoplasm (1)/34%/2,628 patients

2019

3,591 patients

Tumors and neoplasm (1)/35%/62 patients

Acute Lymphoblastic Leukemia

(1)/11.2%/915 patients
Malignant Tumor (2)/2.8%/273 patients

Tumors and neoplasm (1)/32.1%/2,341 patients

2021

2,386 patients

Tumors and neoplasm (1)/34.2%/51 patients

Acute Lymphoblastic Leukemia

(1)/12.5%/670 patients
Malignant Tumor (2)/1.3%/68 patients

Tumors and neoplasm (1)/29.7%/1,597 patients

Position among the main causes appear in parenthesis/Percentage from total/Number of patients

  1. Specify please at which one of the National Institutes the data were collected.

R: The revised Hospital’s name in “Section 2.1 Design” is now presented as follows:

“ …10 medical performance indicator results to evaluate the performance of the Hospital Infantil de Mexico Federico Gómez which is one of the National Institutes of Health in Mexico”.

  1. Which specialists collected and analysed the data?

R: “Data were collected and analysed by the authors of this paper..”

  1. Line 293: Drop “and existence of results in the analyzed period” (as it is obvious from line 292).

R: The line was droped.

  1. Line 296: Change “3 models” to “3 models (ES, ARIMA and LR)”

R: Line change to: 3 models were evaluated (ARIMA, LR and ES).

  1. Lines 296-297: These two sentences belong rather to Procedure-section. (One of them is already there – see line 304).

R: The two sentences were moved to procedure section.

  1. Lines 298-299: This sentence belongs rather to Statistical Analysis section

R: The sentences were moved to statistical analysis section.

Procedure:

  1. Line 300: Procedure for what?

R: The line change to say “Procedure for measuring the performance of each model”.

  1. The sentence starting with “We made ..” can be shortened to “We made quarterly forecasts of performance indicator results with four quarterly lagged results as a predictor”.

R: The sentence change to say “We made quarterly forecasts of performance indicator results with four quarterly lagged results as a predictor” as the reviewer recommended.

  1. Line 307: Drop “of the 3 statistical predictive models” (as it is obvious from the previous text).

R: The line was droped.

Statistical Analysis:

  1. You don’t need to mention so many times that significance level chosen was 5%. Just mention it once.

R: The sentence change to say “We made quarterly forecasts of performance indicator results with four quarterly lagged results as a predictor” as the reviewer recommended.

  1. Which statistical tests were used to derive p-values?

R: The sentences were modified to provide specific details for ARIMA, LR, and ES, aand now is indicated in the manuscript as follow.

A value of p < 0.05 is used to provides evidence to reject the null hypothesis in favor of the alternative hypothesis. The ARIMA Dickey-Fuller test was employed with the null hypothesis that time series data has a unit root equal to 1 or it is non-stationary. The alternative hypothesis asserts that the root differs from 1, signifying the absence of a unit root and ensuring that the data is stationary with a stable mean and variance. Addition-ally, the MacKinnon test was utilized for its enhanced precision in critical values, partic-ularly for small sample sizes. This test represents an improved iteration of the Dick-ey-Fuller test and aligns with the aforementioned logic.

For LR model the null hypothesis is that the coefficient of the independent variable (time) is equal to zero, meaning that do not have a significant effect on the dependent var-iable (performance indicator results). The alternative hypothesis, in contrast, asserts that there is a significant linear relationship between the variables. A significant p-value in linear regression means that there is sufficient evidence to reject the null hypothesis and suggests that there is a relationship between the independent variable and the dependent variable.

For ES model, the “desired responsiveness index or optimal factor α” was calculated to obtain the predicted values. ES is not directly associated with a p-value as in statistical hypothesis testing. Instead, exponential smoothing is a forecasting method that uses a weighted averaging approach to predict future values in a time series.

  1. Lines 472-474: Drop as this information has already been given earlier.

R: The lines were droped.

  1. Line 474: Change “number” to “numbers”

R: The word “number” cahnged to “numbers”.

  1. Lines 479-481: Drop as it has been already explained in the Procedure-section.

R: The lines were droped.

  1. Line 482: Not clear, what results.

R: The redaction was changed to:

For the first specific objective, the MAE result for each model was used to identify the predictive model with the smallest error between the real and predictive values.

  1. Line 483: Not clear, variation of what.

R: The redaction was changed to:

The smallest standard deviation also was considered as an indication of data consistency and homogeneity, as it resulted in better predictive values in relation to the mean.

  1. Line 482, regarding p-value: What was the null hypothesis?

R: Please review the response provided above in number 28.

  1. Line 488: Change “The Dockey-Fuller test” to “For ARIMA method, the Dockey-Fuller test“.

R: The modification was implemented as follows:

“For ARIMA method, the Dickey-Fuller test“

  1. Line 487: Drop (see comment 35)

R: The line was droped.

  1. Line 508, regarding p-value: What was the null hypothesis?

R: Please review the response provided above in number 28.

  1. Line 515, regarding p-value: What was the null hypothesis?

R: Sentences changed to:

For the third specific objective to prove the validity of the RBM in relation to the per-formance indicator results analysis with predictive models, the best model can be used in order to calculate future expected values.

  1. Line 516: standard deviation for what?

R: Same as in 38; Sentences changed to:

For the third specific objective to prove the validity of the RBM in relation to the per-formance indicator results analysis with predictive models, the best model can be used in order to calculate future expected values.

Results:

  1. Table 1: Add total numbers.

R: Table 1 now shows  the total number of patients, see response to number 17.

  1. Table 2: Move performance indicators titles to the left, otherwise it is difficult to read the table.

R: Table 2 has been enhanced to incorporate performance indicator titles within an extra left-hand column; similarly, table 3 has also been adjusted to encompass the same suggested alteration.

  1. Table 2: “*” is used in column “P Value” without explanation in the legend.

R: At the end of table 2, an explanatory note regarding the p-value has been appended, stating as follow:

Abbreviations: LR Linear regression, ES Exponential Smoothing, MAE: mean absolute error; NA: does not apply; p < 0.05 there is evidence to reject the null hypothesis.

  1. Line 533: Wrong numbering. Change the section number to 3.3 (with corresponding change in numbering of further sections)

R: The section numbers have been rectified to 3.3, 3.3.1, 3.3.2, and 3.3.3.

  1. Table 3, legend: what is “movil”?

R: The word movil have been rectified to moving.

  1. Line 557: Refer to a table.

R: Table 2 has been added and enclosed within parentheses as a reference.

  1. Line 558: Table 3 doesn’t include any results for linear regression.

R: Table 3 has been updated to Table 2.

  1. Line 560 and Table 4: “validation” is a wrong term here. Using these methods, you don’t validate LR, but check its assumptions.

R: The terms were altered as follows:

“The assumptions were checked as shown in Table 4.”

  1. Table 4: Very confusing. If it is about LR (as the title states), why other methods are mentioned in column 1?

R: The methods listed in column 1 have been removed.

  1. Table 4: Not clear, what mathematical terms the numbers in columns 2-4 represent. If p-values, then its interpretation in lines 567-573 is wrong (see comment 11).

R: The interpretation has been rectified as outlined below:

Seven performance indicators demonstrate normality as per the Shapiro-Wilk test. However, the first two indicators and the percentage of emergency admissions do not exhibit a normal distribution. It's important to note that since p > 0.05, we do not reject the null hypothesis suggesting that the data follows a normal distribution. We have effectively addressed outliers; within the 20 data points analyzed, three indica-tors have one or two data points exceeding the threshold. This indicates that the simple linear regression model is not significantly affected by unusual values. Lastly, the Breusch-Pagan test reveals that nine out of the ten indicators show homoskedasticity, confirming the accurate estimation of coefficient errors. Similarly, as with the Shapiro-Wilk test, the p-value > 0.05 supports that null hypothesis coul not be rejected suggesting homoskedasticity.

  1. Section 3.2.3: Refer to a table.

R: Table 2 has been included and placed within parentheses as a reference. Section 3.2.3 has been corrected to 3.3.3 (refer to number 43).

Discussion:

  1. Line 614: What is NIH?

R: was modified to: “at the Hospital Infantil de Mexico Federico Gomez”.

  1. Lines 615-819: Statements regarding validation, accuracy and reliability are wrong. The tests mentioned were used only to check the methods assumptions.

R: was modified to:

To check methods assumptions of the LR model, we employed the Shapiro-Wilk, Cook's distance, and Breusch-Pagan tests. Our results demonstrate that the model fits appropriately for the majority of the performance indicators.

  1. Limitations: Discuss limitations regarding using p-values for decision-making. E.g., dependence of p values on sample-size: for large samples p-values can be small even if difference doesn’t exist and the other way around.

R: was modified to:

It's worth noting that the study has several limitations. Firstly, the absence of probabilistic sampling impacts the generalizability of findings. Secondly, the utilization of p-values for decision-making warrants caution due to their sensitivity to sample size; in large samples, p-values may appear small even when differences are absent, and vice versa. Additionally, minor fluctuations in data or analytical methodologies can lead to substantial variations in p-values, influencing result interpretations.

  1. Limitations: Discuss limitations of statistical tests you used to check assumptions of LR.

R: was modified to:

Moreover, the tests chosen to assess LR assumptions (Shapiro-Wilk, Cook's dis-tance, and Breusch-Pagan tests) might not be the most suitable options and could ben-efit from the inclusion of Q-Q plots. Tests relying on p-values can sometimes detect in-significant differences as significant in larger sample sizes, while the reverse problem arises for smaller samples. Notably, there are other statistical tests available that were not utilized to assess LR assumptions, such as assessing independence of errors, de-tecting multicollinearity, identifying autocorrelation, and verifying constant variance.

  1. Line 679: What do you mean under “automated tool”? (Is Stata not “automated” enough compared to Phyton and R?

R: was modified to:

To use an automated tool to collect, analyze and present performance indicator results such as Python or R which are programming languages are versatile tools that go beyond statistical analysis, encompassing web development, automation, and machine learning capabilities. Additionally, they provide robust data visualization options through librar-ies, enhancing data exploration and presentation. Moreover, their open-source nature en-sures widespread accessibility, making them cost-effective alternatives to STAT.

Conclusions

We greatly value your feedback and the errors you've pointed out, our aim was to provide a detailed description of three statistical methods to illustrate their use in helping administrators, professionals, and decision-makers convert data into valuable information for making informed decisions. We also aimed to identify which method best fits the real expected values, thus providing statistical evidence to analyze cancer data statistics.

The central contribution of our work lies in demonstrating how well the described instrument predicts trends, enabling corrective actions and improving the effectiveness and quality of medical services delivered to children with cancer. The results underscore the importance of implementing performance indicators associated with cancer and analyzing them with predictive models for the monitoring and evaluation of cancer cases.

The modifications carried out have reinforced the key points that underscore the significance of this study, serving as a rationale for the utilization of predictive models, especially within the unique landscape of oncology. These adjustments further emphasize the role of predictive models in improving decision-making and optimizing resource allocation in the realm of cancer care.

The above justifies that our work deserves to be published in the prestigious journal "Cancer". 

Reviewer 3

Comments and Suggestions for Authors

I think the authors provided a good revision and the study deserves publication

R: Thank you very much for your approval.

Round 3

Reviewer 2 Report

-